



# A global perspective on $CO_2$ satellite observations in high AOD conditions

Timo H. Virtanen[1], Anu-Maija Sundström[2], Elli Suhonen[2], Antti Lipponen[1], Antti Arola[1], Christopher O'Dell[3], Robert R. Nelson[4], and Hannakaisa Lindqvist[2]

[1]Finnish Meteorological Institute, Climate Research Programme, Helsinki, Finland
[2]Finnish Meteorological Institute, Space and Earth Observation Centre, Helsinki, Finland
[3]Colorado State University, Fort Collins, CO, USA
[4]Jet Propulsion Laboratory, California Institute of Technology, USA

**Correspondence:** Timo H. Virtanen (timo.h.virtanen@fmi.fi)

**Abstract.** Satellite-based observations of carbon dioxide ($CO_2$) are sensitive to all processes that affect the propagation of radiation in the atmosphere, including scattering and absorption by atmospheric aerosols. Therefore, accurate retrievals of column-averaged $CO_2$ ($XCO_2$) benefit from detailed information on the aerosol conditions. This is particularly relevant for future missions focusing on observing anthropogenic $CO_2$ emissions, such as the Copernicus Anthropogenic $CO_2$ Monitoring

mission (CO2M). To fully prepare for CO2M observations, it is informative to investigate existing observations in addition to other approaches. Our focus here is on observations from the NASA Orbiting Carbon Observatory -2 (OCO-2) mission. In the operational full-physics $XCO_2$ retrieval used to generate OCO-2 level 2 products, the aerosol properties are known to have high uncertainty but their main objective is to facilitate $CO_2$ retrievals. We evaluate the OCO-2 product from the point of view of aerosols by comparing the OCO-2 retrieved aerosol properties to collocated Moderate Resolution Imaging Spectro-radiometer

(MODIS) Aqua Dark Target aerosol products. We find that there is a systematic difference between the aerosol optical depth (AOD, $\tau$) values retrieved by the two instruments, such that $\tau_{OCO-2} \sim 0.4\tau_{MODIS}$. We also find a dependence of the $XCO_2$ on the AOD difference, indicating an aerosol-induced effect in the $XCO_2$ retrieval. In addition, we find a weak but statistically significant correlation between MODIS AOD and $XCO_2$, which can be partly explained by natural covariance and co-emission of aerosols and $CO_2$ but is partly masked by the aerosol-induced $XCO_2$ bias. Furthermore, we find that issues in the OCO-2

aerosol retrieval may lead to misclassification of the quality flag for a small fraction of OCO-2 retrievals. Based on MODIS data, 4.1% of low AOD cases are incorrectly classified as high AOD (low quality) pixels, while 16.5% of high AOD cases are erroneously classified as low AOD (high quality) pixels. Finally, we investigate the effect of an AOD threshold on the fraction of acceptable $XCO_2$ data. We find that relaxing the MODIS AOD threshold from 0.2 to 0.5 (at 550 nm), which is the goal for the CO2M, increases the fraction of acceptable data by 14 percentage points globally, and by 31 percentage points for urban

areas.





# 1  Introduction

Anthropogenic emissions of carbon dioxide ($CO_2$) will be monitored operationally in this decade using atmospheric measurements to support the Global Stocktake and provide independent information for tracking national emission reductions outlined in the Paris Agreement (Janssens-Maenhout et al., 2020). An essential monitoring component will be the Copernicus

Anthropogenic $CO_2$ Monitoring Mission (Meijer et al., 2023). While ground-based greenhouse gas measurements are mainly available in developed countries – with limited coverage and representativeness – satellite-based $XCO_2$ information will be irreplaceable in areas where ground-based measurements are not made. The key purpose of the observations is to provide means for an independent verification of nationally reported emissions and, therefore, the focus and the challenge of the CO2M will be in the need to make accurate and precise observations of anthropogenically polluted environments.

The existing satellite $XCO_2$ products from JAXA's Greenhouse Gases Observing Satellite (GOSAT; (Yokota et al., 2009)), NASA's Orbiting Carbon Observatory-2 (OCO-2; (Crisp et al., 2004)), and the Chinese TanSat (Yang et al., 2018) are focused on global $CO_2$ observations and have been developed to inform flux inversion models for quantifying the large-scale sources and sinks of $CO_2$ (e.g., (Houweling et al., 2015; Crowell et al., 2019)). In assimilating satellite data to inverse model systems, the reliability of data has been preferred at the cost of not achieving full global coverage; thus, the observations of potentially

deteriorated quality are filtered in the postprocessing. One of the known factors affecting $XCO_2$ retrieval accuracy and precision are atmospheric aerosols: scattering and absorption by aerosols affects the light path of radiation and complicate the interpretation of the signal (Butz et al., 2009; Guerlet et al., 2013; Connor et al., 2016; Lamminpää et al., 2019; Rusli et al., 2021). Therefore, retrievals made in aerosol-loaded conditions are mostly filtered out (e.g., (O'Dell et al., 2018)). In the advent of CO2M and other missions targeting anthropogenic signals, the focus of flux estimation is shifting from using satellite

data from pristine, aerosol-free scenes to the need to also observe aerosol-contaminated, polluted atmospheres. The goal is to enable reliable quantification of local and regional anthropogenic $CO_2$ emissions, but this poses new challenges to the satellite retrievals.

In the NASA Atmospheric CO2 Observations from Space (ACOS) retrieval algorithm for OCO-2 observations, the aerosol properties are retrieved as part of the full-physics retrieval, and are known to have high uncertainties, in particular for high

aerosol loads (O'Dell et al., 2018). The potential to improve the co-retrieval of aerosols and $XCO_2$ has been emphasized in recent studies (Lamminpää et al., 2019; Sanghavi et al., 2020). A systematic, statistical study on the long data record of OCO-2 observations in quantified aerosol conditions can increase understanding of the potential aerosol effects on $CO_2$ retrievals and support preparations towards the CO2M observations. Reliable information of atmospheric aerosols can be obtained from ground-based instruments and from satellite-based instruments (and algorithms) specialized to detect aerosols, such as Moderate Resolution Imaging Spectro-radiometer (MODIS, Levy and Hsu (2015)). In the latter case, the favorable

orbital configuration of OCO-2 and Aqua satellites as part of the Afternoon-train constellation ensures optimal coverage for collocated observations. This enables an expansion of the evaluation beyond the traditional approaches that are centered around ground-based validation sites (e.g., the Total Carbon Column Observing Network; TCCON Wunch et al. (2011)) from which only a small fraction represent an urban environment.



In this paper, we evaluate the OCO-2 level 2 product from the point of view of aerosols by comparing the OCO-2 estimated aerosol properties to the MODIS/Aqua Dark Target aerosol product. We study how well the current ACOS quality filtering works in different aerosol conditions, focusing in particular on heavy aerosol conditions and urban environments. The focus of this paper is on the statistical analysis of a global, multiyear dataset. For complementarity, we will also use all available TCCON data as a subset of the study and to estimate aerosol and $CO_2$ co-emission.

## 2  Data

### 2.1  OCO-2

NASA's Orbiting Carbon Observatory -2 (OCO-2) is an atmospheric carbon dioxide ($CO_2$) observing mission with a diffraction-grating spectrometer onboard a polar-orbiting satellite. OCO-2 makes passive observations of backscattered solar radiation in the near- and shortwave infrared wavelengths. It has a ground-pixel size of approximately 1 km x 2 km, and covers a swath width of 10 km, with a 16-day revisit time.

We use OCO-2 daily Lite files (V10r) (OCO-2 Science Team et al., 2020), produced by the OCO-2 project at the Jet Propulsion Laboratory, California Institute of Technology, and obtained from the OCO-2 data archive maintained at the NASA Goddard Earth Science Data and Information Services Center (O'Dell et al., 2018; Wunch et al., 2017; Taylor et al., 2023). In the collocated database we include only a limited selection of OCO-2 data fields, but the sounding ID in the combined daily files is equivalent with the original OCO-2 lite files, allowing addition of more data fields in an effective manner. The aerosol parameters of the ACOS algorithm include five scatterers, two cloud types (water and ice), two tropospheric aerosol types and a stratospheric aerosol type (sulfate). Two representative types of tropospheric aerosols (dust, smoke, sulfate aerosol, organic carbon, and black carbon) are drawn from a climatology based on location and time (Crisp et al., 2021). From the large number of quantities provided by the Atmospheric Carbon Observations from Space (ACOS) Level 2 full-physics (L2FP) retrieval algorithm, we use mainly the estimates of the $CO_2$ column-averaged dry-air mole fraction ($XCO_2$), the total aerosol optical depth (AOD) values, and the $XCO_2$ quality flag.

### 2.2  MODIS

We use the level-2 (L2) Moderate Resolution Imaging Spectro-radiometer (MODIS) Collection 6.1 atmospheric aerosol product from the Aqua platform (MYD04_L2) as reference aerosol data (Levy and Hsu, 2015). The MODIS Dark Target (DT) algorithm (Levy et al., 2013) is available over ocean and dark (e.g., vegetated) land surfaces, while the MODIS Deep Blue (DB) (Hsu et al., 2004) covers land areas including bright surfaces. As we are mainly interested on the effect of aerosols on $XCO_2$ over urban areas, we concentrate on MODIS retrievals over land surfaces and use mainly the 10 km MODIS DT product over land; results for DB are shown in Appendix A. Both Aqua and OCO-2 are in the Afternoon-train satellite constellation following similar orbital tracks allowing fair collocation between the instruments. MODIS data used in this study were obtained from the NASA Level-1 and Atmosphere Archive & Distribution System Distributed Active Archive Center (LAADS



DAAC) https://ladsweb.modaps.eosdis.nasa.gov/. Five years of data from 2015 to 2019 were processed. Due to the large size of the original MODIS L2 aerosol data, the data were pre-processed before collocating with OCO-2 data to create daily files which contain a reduced number of original data fields and cloud-screened pixels only. MODIS quality flag was applied to remove the poor quality pixels (MODIS quality flag 0).

### 2.3 TCCON

For ground-based reference $XCO_2$ measurements, we employ the Total Carbon Column Observing Network (TCCON) which consists of high-resolution Fourier Transform Spectrometers that make observations of direct sunlight in the near-infrared wavelengths. TCCON provides precise and accurate retrievals of the total column $CO_2$ abundance (Wunch et al., 2011). In this study, we use data from 26 TCCON stations to quantify the AOD dependence of $XCO_2$ (Table A6).

### 2.4 AERONET

The AErosol RObotic NETwork (AERONET) is used as ground based reference data for aerosol optical depth (AOD). AERONET is a network of over 600 stations (currently) using standardised methodology and equipment to measure aerosol optical, microphysical, and radiative properties (Holben et al., 1998). The AERONET sunphotometer measurements are routinely used as reference measuremetns for satellite aerosol retrievals due to their high accuracy ($\sim$ 0.01-0.02, Eck et al. (1999)). In this work we use AERONET Version 3 level 2.0 data at 500, 675 and 870 nm to evaluate the OCO-2 total AOD. We consider AERONET data collocated with OCO-2 glint and nadir observations for September 2014 - February 2023.

## 3 Methods

### 3.1 Collocation of MODIS and OCO-2 data

The OCO-2 and MODIS data are collocated using the OCO-2 daily (lite) files and reduced daily MODIS files. The reduced daily MODIS files are first created starting from the L2 MODIS Aqua AOD files (MYD04) over land areas by removing pixels that do not have valid aerosol retrieval results and by dropping some data fields which are not relevant for this study. The collocation is done by selecting the nearest MODIS pixel for each OCO-2 pixel within a $0.2° \times 0.2°$ area and within one hour of OCO-2 overpass (to remove possible overlapping orbits of the same day at high latitudes).

To further reduce the data size, the collocated dataset contains only selected OCO-2 and MODIS data fields, but it is straightforward to add other OCO-2 data fields later if necessary. Also, the collocated dataset includes only those OCO-2 data points for which a MODIS match is found. This reduces the number of data points to about 14% of the original OCO-2 data points for the five years considered (2015-2019). Table A1 shows the number of original OCO-2 data points and the number of collocated data points with MODIS match for each year (2015-2019). Using the MODIS DT-land retrieval removes oceans and bright surfaces such as deserts and snow covered areas, and the MODIS cloud mask may further reduce the number of data. This reduces the coverage of the collocated dataset with respect to the original OCO-2 data especially at high latitudes. Fig. A1





shows the fraction of OCO-2 pixels with a MODIS match for $1° \times 1°$ grid cells, and the fraction of good quality pixels (OCO-2 quality flag) for the collocated data. The collocated dataset in netcdf format is available as open data (Virtanen, 2024).

### 3.2   Collocation with TCCON

OCO-2 v10 $XCO_2$ observations were collocated with TCCON using the following criteria. Spatially, all satellite observations

within 1 degrees in latitude and 1.5 degrees in longitude from a given TCCON site were collected and, for each observation, a corresponding TCCON $XCO_2$ value was assigned as the mean of TCCON $XCO_2$ retrievals within $\pm 60$ minutes from each OCO-2 observation. The effect of different prior profiles in OCO-2 v10 and TCCON GGG2020 was taken into account by adjusting the OCO-2 $XCO_2$ value, following Mendonca et al. (2021). In practice, this adjustment was very small, given the similarity of the prior profiles. The different vertical sensitivity of the TCCON and OCO-2 retrievals was taken into account

by adjusting the retrieved, collocated TCCON $XCO_2$ values (Mendonca et al., 2021).

### 3.3   Collocation with AERONET

Each nadir or glint mode OCO-2 observation close to an AERONET site is matched with the ground based observations using the following criteria. Spatial collocation uses distance threshold of $0.1°$ around all available AERONET sites and temporal collocation averages AERONET observation within $\pm 30$ minutes of the satellite overpass. The OCO-2 observations are not

spatially averaged. A simple average of AOD values at 675 and 870 nm is used to evaluate the effect of wavelength difference (see Fig. A5).

### 3.4   Aggregation of collocated data

Analysing the collocated data of this size ($\sim 10$ M points for a year) requires some aggregation before plotting. Two approaches have been applied: **1)** Data fields are aggregated to an AOD vs. AOD grid, i.e. data points falling in certain MODIS AOD bin

and certain OCO-2 AOD bin are averaged (e.g. Fig. 5). For MODIS, we use AOD at 550 nm and for OCO-2 the total AOD data field. The number of data points in each AOD matrix grid cell is also recorded (e.g. Fig. 2). **2)** In the second approach the data is aggregated to a regular lat/lon grid (e.g. Fig. 1). Optionally, the OCO-2 quality flag (QF) can be applied in the aggregation, removing low quality pixels. Aggregation is done using all available collocated data over five years (2015-2019). Several subsets of the collocated data set are also analysed respectively, including different years, different seasons (combined

over all years), and different geographic areas (Table A2).

### 3.5   Linear trend correction for $XCO_2$

For the multiyear dataset we use a simple detrending of OCO-2 $XCO_2$ values to compensate for the steady increase of CO2 levels in order to focus more on the details of $XCO_2$ variability and possible retrieval biases. A reference date is set at 1 January 2015, and a linear increase of 2.4ppm/y is assumed and corrected for in the data (ten year global average, NOAA

Global Monitoring Laboratory (last access: 22 April 2024)). We call this process the linear trend correction, and when applied



to the $XCO_2$ data in this work, we denote this by the abbreviation LTC. While this approach allows meaningful aggregation of $XCO_2$ data over several years, it does not take into account the (spatially varying) seasonal variation of $XCO_2$.

### 3.6 $XCO_2$ anomaly

The OCO-2 $XCO_2$ anomaly is calculated for each good quality OCO-2 pixel in the collocated dataset as the difference from the median $XCO_2$ value calculated within 500 km for the corresponding OCO-2 orbit. This is an alternative way to de-trend the data, instead of applying the simple LTC. Unlike LTC, the anomaly method also effectively de-seasonalises the data. It also allows to study the covariance of AOD values and local $XCO_2$ anomalies caused by possible CO2 sources and sinks.

## 4 Results

In this section we discuss relations and implications of five years of global collocated MODIS and OCO-2 data. We will first consider the differences between collocated MODIS and OCO-2 AOD data, and then proceed to analyse the connection between these AOD differences and the $XCO_2$ values retrieved with OCO-2. The goal here is not to focus on individual retrievals, but rather to explore statistical connections on global five-year dataset. Besides the full dataset, we will consider the effect of OCO-2 quality filtering, the effect of various AOD thresholds for 'good quality data' and the effect of splitting the data into different subsets, such as urban areas which will be particularly relevant for CO2M. We will also address seasonal and annual variability, as well as spatial variation. In this context, in order to remove the effect of increasing $XCO_2$ values over the five years, we apply the simple linear trend correction (LTC) described in the Methods section. As an alternative de-trending option for $XCO_2$ we use anomaly data (see Methods), which is useful in removing also the seasonal effect, preserving ideally local-scale spatial variability. Finally, we use ground-based $XCO_2$ reference data from TCCON, collocated with both MODIS and OCO-2, to study the effect of aerosols to the $XCO_2$ retrievals in a statistical sense.

### 4.1 AOD comparison

Figure 1 a) shows MODIS DT AOD at 550 nm aggregated to a $1° \times 1°$ lat/lon grid for 2015-2019 for quality-filtered collocated data (using the OCO-2 quality flag) over land. High AOD areas due to anthropogenic aerosol emissions are seen in particular in parts of Asia, biomass burning aerosols increase AOD in central Africa and South-East Asia, and elevated aerosol loads due to dust are seen over various desert areas around the globe. MODIS Dark Target observations are not available over bright surfaces such as large deserts and snow-covered areas, which explains the gaps seen on the map. Figure 1 b) shows the AOD difference between OCO-2 and MODIS. The largest differences in AOD appear to be concentrated largely in the high AOD areas in South-East Asia, where OCO-2 AOD is lower than MODIS AOD. Also, for several areas with low MODIS AOD, OCO-2 shows higher values (positive AOD difference), e.g. in parts of Brazil and Australia. These positive difference values are related to the MODIS DT algorithm permitting small negative AOD values. In short, the negative values mean that the AOD is low, but the exact value is not certain. While the negative values are unphysical, they are kept in the data in order to avoid a positive bias in the data (Sayer et al., 2014). The AOD difference is also positive for the Sahel region where the MODIS





DT values in the collocated dataset are low. The Sahel area is known to have occasional high AOD caused by desert dust. Part of these cases are removed by the OCO-2 quality filtering. MODIS DT algorithm has lower AOD values compared to MODIS Deep Blue algorithm in this region. The AOD map and AOD difference map for MODIS DB are shown in the Fig. A2. We see

that MODIS DB shows higher AOD than OCO-2 more often than MODIS DT.

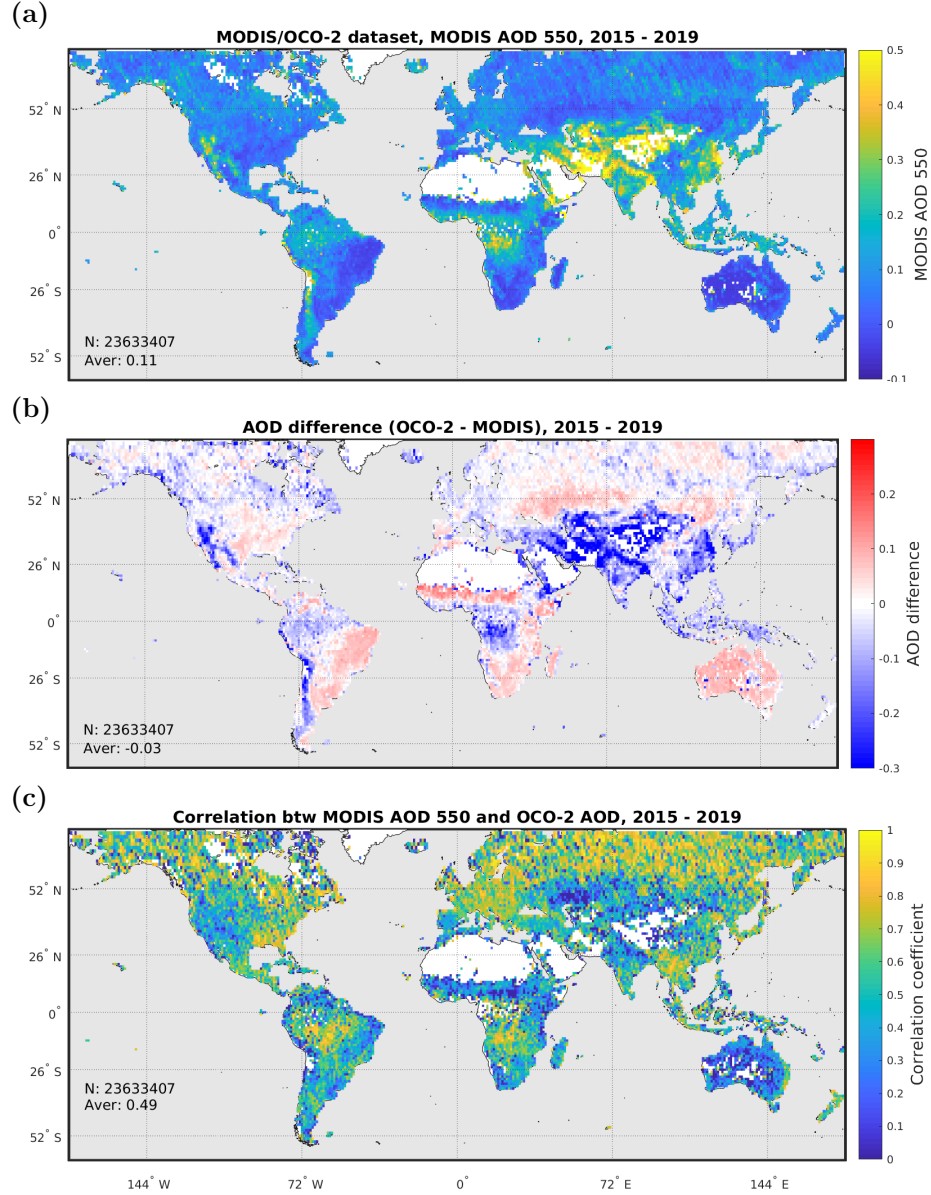

**Figure 1.** Collocated OCO-2 v10 and MODIS Aqua DT-land dataset five year $1° \times 1°$ aggregate maps for quality filtered data. **a)** MODIS AOD at 550 nm. **b)** AOD difference (OCO-2 - MODIS). **c)** Correlation between MODIS and OCO-2 AOD values for $1° \times 1°$ grid cells.





Figure 1 c) shows Pearson correlation coefficient R between MODIS AOD at 550 nm and OCO-2 total AOD for $1° \times 1°$ grid cells for five years. The data is rather noisy, but regions with particularly low correlation are seen, including Australia, Sahel, Western USA and the arid regions of Central Asia. Good correlation is observed in Europe, northern high latitudes, and over tropical rainforests. MODIS DB shows roughly similar patterns. Figure A3 a) shows a global timeseries comparison
for MODIS and OCO-2 AODs. The correlation coefficient calculated from monthly temporal bins (R=0.53) is similar to the average spatial correlation in Fig. 1 c) (R=0.49).

Here we point out that the MODIS AOD is evaluated at 550 nm wavelength, while the OCO-2 total AOD value corresponds to 755 nm, and the two are hence not directly comparable. We do not expect to see a one-to-one correspondence between the two. The sensitivity of AOD on the wavelength depends on the aerosol size distribution and other properties. In general, for
typical ambient aerosols, it is expected that the AOD is smaller at 755 nm, as suggested by the data. One way to scale the AOD obtained at one wavelength to other wavelength is to use the Ångström exponent. While MODIS-based estimates of Ångström exponent exist, they are not reliable over land (Levy et al., 2010). To obtain a rough idea about how the wavelength difference might affect the AOD comparison on global scale, we have used the Ångström exponent from collocated MERRA-2 monthly climatology (Global Modeling And Assimilation Office and Pawson, 2015) to scale the OCO-2 AOD values to 550 nm, which
can be considered as a reference wavelength used in many satellite aerosol products. The result suggests that the low bias in OCO-2 AOD compared to MODIS is only slightly reduced by the scaling (Fig. A4). A bivariate linear fit for OCO-2 AOD (at 755 nm) as function of MODIS AOD (at 550 nm) gives a slope 0.3, while a fit using OCO-2 AOD scaled to 550 nm gives a slope 0.4 (without the OCO-2 quality filtering).

Comparison of OCO-2 AOD with AERONET shows similar results (Fig. A5). A linear fit of OCO-2 AOD against AERONET
AOD at 500 nm gives a slope 0.3, while a fit against AERONET AOD scaled to 770 nm gives a slope 0.53. The slope is further increased when a more recent version of OCO-2 algorithm is used. The similarity of these results supports the assumption that MODIS AOD can be used as reference data in evaluating the OCO-2 performance. While MODIS AOD product certainly has higher uncertainty than AERONET, it expands the comparison to a global scale.

The OCO-2 quality filtering applied to the collocated data set heavily affects the AOD comparison shown in Figure 1.
Because the cases where OCO-2 retrieves large AODs are removed by the quality filtering, the aggregated MODIS DT AOD values are much lower than they would be for unfiltered MODIS data. The quality filtering also causes a sampling bias between MODIS and OCO-2 AOD data, since not all cases with high MODIS AOD are removed. Statistics for AOD in different subsets are shown in Tables A1 to A4. The correlation is better for unfiltered data (Table A4). Also, the correlations are slightly better (0.59) in summer (JJA) than in other seasons. The correlation is particularly poor in Australia (0.22), for DJF in North Asia
(-0.39, very few data), and for DJF in South-America (0.09). Highest MODIS AOD values are observed in urban areas and in South-East Asia. The AOD differences are also largest in these areas, as the OCO-2 AOD are less pronounced in these regions.

Finally, we note that the OCO-2 retrieval algorithm ACOS is not an aerosol retrieval algorithm and the total AOD value included in the product is only one of more than fifty components in the full physics retrieval. Incorrect AOD values in the ACOS retrieval may be compensated by other retrieval parameters, and a difference between MODIS and OCO-2 AOD values
does not necessarily indicate erroneous $XCO_2$ retrieval. Our focus here is not to evaluate the AOD component of ACOS



retrieval as such, but to study the statistical relationships using MODIS AOD as independent reference data. We also note that the collocation between MODIS 10 km AOD product and the OCO-2 observations at higher spatial resolution affects the comparison. The collocation approach applied here, using the closest MODIS pixel for each OCO-2 data point, is the simplest possible. The simple approach was chosen to enable processing the large dataset efficiently, and more sophisticated collocation for detailed case studies are considered elsewhere.

The expected error envelope for MODIS DT AOD is $\pm 0.05 + 0.15\tau_{\mathrm{A}}$ for reference (AERONET) AOD $\tau_{\mathrm{A}}$ (Levy et al., 2010, 2013), indicating the relatively high uncertainty at low AOD values. However, the absolute value of AOD (or the absolute difference between MODIS and OCO-2) at the very low levels is not crucial for the accuracy of the $XCO_2$ retrieval, since the effect of aerosols is expected to be small for low AODs. In addition, the cases where OCO-2 severely overestimated AOD are not seen in the quality filtered dataset, as the cases with OCO-2 AOD over 0.2 are removed by the standard quality filtering (O'Dell et al., 2018). Hence, from the point of view of the aerosol effect on the $XCO_2$ retrievals, the most important areas are those with AOD difference below -0.2 (blue areas in Fig. 1 b), where OCO-2 AOD is significantly lower than MODIS AOD. In the following we will separate the data into different AOD difference subsets to study this in more detail.

## 4.2 AOD matrices

Figure 2 shows joint histograms of five years of collocated OCO-2 and MODIS AOD data (over 40 million collocated data points). In panel a) we show all data, without OCO-2 quality filtering. In panel b) we have applied filtering using the OCO-2 quality flag (O'Dell et al., 2018), which identifies potentially bad quality retrievals affected by, e.g., clouds or high aerosol loads, and removes the results with OCO-2 AOD higher than 0.2. The dashed red line shows bin-averaged OCO-2 AOD data for MODIS AOD bins (fifty bins with width 0.02). We see that OCO-2 AOD is systematically low with respect to MODIS AOD (mean MODIS AOD is 0.15, mean OCO-2 AOD is 0.12), except for the lowest MODIS AOD values where OCO-2 has higher AOD. The overestimation at the low AOD end may be caused by the water and ice aerosol components included in the OCO-2 total AOD. The dashed green line shows a bivariate linear fit, which follows closely the binned mean values with a slope 0.33 for the unfiltered data. Naturally, the quality filtering causes deviation from the linear behaviour and a lower slope (0.18) for the linear fit, since the high MODIS AOD values are not removed by the filtering.

Pearson correlation coefficient for the unfiltered data is 0.60, reducing to 0.52 for the filtered data, which indicates that the sampling is biased for the quality filtered data (higher MODIS AOD values remain in the collocated data set). Note that in the collocated dataset the MODIS data is often the limiting factor (Table A1), already removing data over bright surfaces and in proximity of clouds. Applying the OCO-2 quality filter further reduces the collocated data to 56% of the original collocated data points. Most, but not all, of this reduction can be contributed to removing the high AOD cases.

The dotted black lines in Fig. 2 at AOD threshold of 0.2 divide the AOD matrix into four quarters Q1-Q4. The threshold 0.2 corresponds roughly to the current limit for good quality retrievals in OCO-2. We note that since the wavelength-corrected linear relation between the two instrument is roughly $AOD_{\mathrm{MODIS}} \sim 2.5\ AOD_{\mathrm{OCO-2}}$, a more appropriate AOD threshold for MODIS could be 0.5. For simplicity we use here the same limit 0.2 for both instruments, but in section 4.5 we study the effect of filtering the data with AOD threshold 0.5 applied to MODIS data. The first quarter Q1 with AOD from both instruments





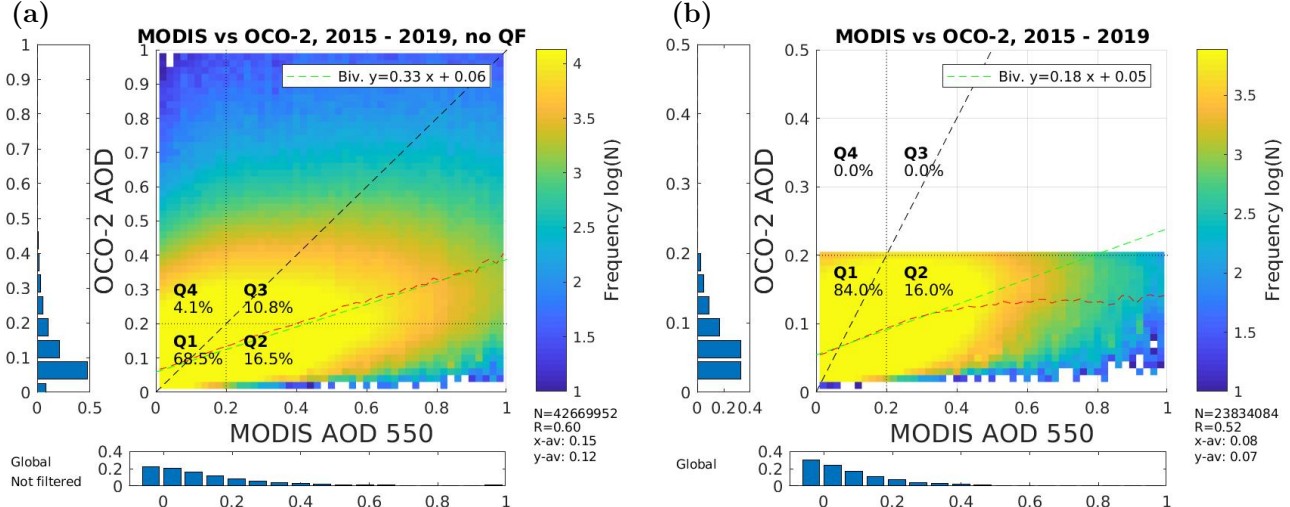

**Figure 2.** Number of collocated data (logarithmic color scale) for each 'AOD grid cell' ($50 \times 50$ cells of AOD width 0.02). Left: all data, right: good quality data only. The dashed red line shows average OCO-2 AOD for each MODIS AOD bin. The dashed green lines shows a bivariate linear fit. The dotted black lines divide the data to four 'AOD-quarters' Q1-Q4 (see text). The text insets show the fraction of data in each quarter. The dashed black line shows the 1:1 line. The normalized AOD histograms show the distribution of data respectively for OCO-2 (left) and MODIS (bottom). The lower right text inset shows the number of data, correlation coefficient (R) and average AOD values for MODIS (x-av) and OCO-2 (y-av), respectively.

below 0.2 contains most of the data (68.5%). The second quarter Q2 contains data with $\tau_{\text{OCO}-2} < 0.2$ and $\tau_{\text{MODIS}} > 0.2$ (16.5%). These data points are assumed to have low AOD in the OCO-2 retrievals, but according to MODIS there can be quite heavy aerosol loads, which might affect the $XCO_2$ retrievals. Q3 contains data points with AOD above 0.2 for both instruments (10.8%). Most of these data points are removed when the OCO-2 quality filtering is applied, which is appropriate considering that heavy aerosol conditions should be avoided in $XCO_2$ retrievals. The last quarter Q4 includes data points for

which $\tau_{\text{OCO}-2} > 0.2$ and $\tau_{\text{MODIS}} < 0.2$ (4.1%). Most of these data are removed by quality filtering, but based on low MODIS AOD values Q4 could contain good quality retrievals.

Table 1 shows the fraction of data in different quarters of the AOD matrix and the total number of data points in the collocated MODIS/OCO-2 dataset and two subsets. The numbers are shown respectively for quality filtered (good quality) and for the unfiltered (all data) cases. The global dataset includes all available OCO-2 data from 2015-2019 which have a matching

MODIS aerosol retrieval (14% of all OCO-2 datapoints, over 40 million datapoints in total, see Table A1). The urban dataset is limited to areas of dense human habitation using the urban area mask from naturalearthdata.com (Ver. 4.1.0) (NaturalEarth, last access: 22 April 2024; Schneider et al., 2009), illustrated by Fig. A7. The OCO-2/TCCON dataset contains collocated MODIS/OCO-2/TCCON data for the 26 TCCON sites listed in Table A6. The fraction of data in Q2 is considerably higher for the urban subset, reflecting the higher AOD differences between the two instruments over urban areas. We see that the quality





filtering using OCO-2 quality flag removes also part of data from Q1 and Q2, while some data points remain in Q3 and Q4. The ~24 million good quality data points for the global dataset compose about 56% of the total collocated data, which is about 66% of data originally in the two lower quarters Q1 and Q2.

| | All data | | | | | Good quality | | | | | |
|---|---|---|---|---|---|---|---|---|---|---|---|
| | Fraction of data (%) | | | | $N_{\mathrm{all}}$ | Fraction of data (%) | | | | $N_{\mathrm{QF}}$ | $N_{\mathrm{QF}}/N_{\mathrm{all}}$ |
| $XCO_2$ dataset: | Q1 | Q2 | Q3 | Q4 | $(\times 10^6)$ | Q1 | Q2 | Q3 | Q4 | $(\times 10^6)$ | (%) |
| Global | 68.5 | 16.5 | 10.8 | 4.1 | 42.7 | 84.0 | 16.0 | 0.0 | 0.0 | 23.8 | 55.9 |
| Urban | 52.9 | 34.2 | 11.5 | 1.5 | 0.9 | 63.8 | 36.2 | 0.0 | 0.0 | 0.5 | 61.1 |
| TCCON | 77.0 | 17.9 | 3.2 | 1.9 | 1.0 | 83.5 | 16.5 | 0.0 | 0.0 | 0.7 | 65.9 |

**Table 1.** Fraction of data in different AOD quarters for different subsets of the collocated MODIS/OCO-2 datasets. 'Global' set includes all collocated data, 'urban' subset is limited to urban areas (see text), and 'TCCON' subset is further collocated with TCCON stations (subset 1 with OCO-2 $XCO_2$ values, subset 2 with TCCON $XCO_2$ values).

An AOD comparison showing OCO-2 AOD as function of MODIS AOD with binned averages and bivariate linear fits for different subsets of the collocated data are shown in Fig. A9 (for quality filtered data) and A10 (without quality filtering). The linear fit slopes and correlation coefficients are summarized in Table A5. The sampling bias caused by OCO-2 quality filtering affects the bin averaged plots for high MODIS AOD values, causing deviation from linear behavior. Most of the collocated data are in the low AOD region, and hence the linear fits are less affected by this bias. For the unfiltered data the binned averages follow the linear fits closely (for most cases). We see that the AOD slopes are very similar for different years and different seasons. For the urban and TCCON subsets the slopes are slightly lower, indicating more pronounced low bias of OCO-2 AOD compared to MODIS. For the different geographic areas there is more variation.

As already noted, the MODIS DT aerosol product contains a considerable fraction (~20%) of negative AOD values. While these are obviously unphysical, they are kept in the analyses in order to not disturb the AOD distribution (Sayer et al., 2014). These data are not shown in Fig. 2, but in the statistics we include the negative MODIS AOD data points to the AOD quarters Q1 and Q4, depending on the corresponding OCO-2 AOD value.

Figure 3 shows maps of the fraction of data in the two AOD matrix quarters Q1 and Q2 for the good quality data per $1° \times 1°$ grid cell. The map for Q1 fraction reveals that for the vast majority of land regions, average AOD is less than 0.2 for both instruments; however, large areas in South East Asia and Central Africa have a low fraction of data in the low AOD quarter, and correspondingly a higher fraction of data in Q2. Therefore, these areas are more sensitive to effects caused by high aerosol loads in the $XCO_2$ retrieval. Fig. A7 shows the fraction of data in Q1 for the urban subset.

### 4.3 Connection between $XCO_2$ and AOD

In this section we consider the possible aerosol effects in the OCO-2 $XCO_2$ retrieval. Figure 4 a) shows aggregated OCO-2 $XCO_2$ values over the globe for the collocated dataset. Visual comparison with the AOD map in Fig. 1 shows some spatial correlation between high AOD and high $XCO_2$ values. This spatial correlation between high $XCO_2$ and high AOD values may



(a)

**Fraction of data, 2015 - 2019, Q1**

(b)

**Fraction of data, 2015 - 2019, Q2**

**Figure 3.** Fraction of data in AOD quarters Q1 (both AODs< 0.2) and Q2 (OCO-2 AOD below 0.2, MODIS AOD above 0.2) for five years of data.

affect the $XCO_2$ statistics in two ways: First, a larger fraction of data is removed by the OCO-2 quality filtering over the high

$XCO_2$ load areas. Second, considering Fig. 3 for the quality filtered data shows that areas with a large fraction of data in AOD quarter Q2 typically have high $XCO_2$ values. These heavy aerosol conditions suggested by MODIS data, which remain in the quality filtered OCO-2 dataset, may affect the $XCO_2$ retrieval quality. Figure 4 b) shows the correlation between MODIS AOD and OCO-2 $XCO_2$ for $1° \times 1°$ grid cells. We see particularly high correlation values for the Sahel region, parts of South-East Asia, and Western USA. Figure A3 b) and c) show global time series of collocated OCO-2/MODIS data, revealing a moderate

(R=0.18) temporal correlation between MODIS AOD and OCO-2 $XCO_2$.

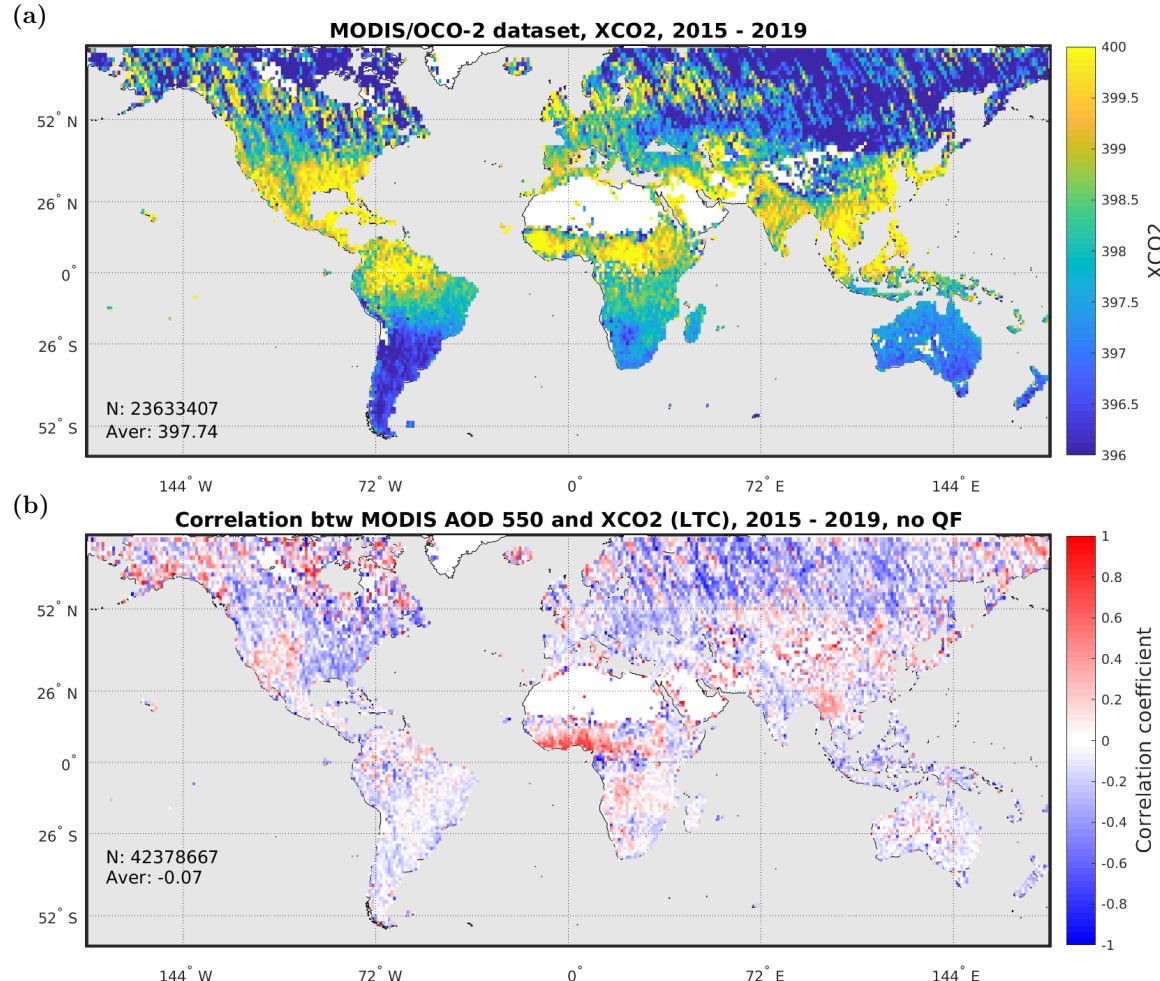

**Figure 4.** Linear-trend-corrected OCO-2 XCO$_2$ data from the collocated OCO-2 and MODIS dataset for five years (2015-2019). **a)** Quality filtered XCO$_2$ (LTC) data aggregated to $1° \times 1°$ grid cells. **b)** Correlation between MODIS AOD and OCO-2 XCO$_2$ for $1° \times 1°$ grid cells (quality filter not applied).

The sampling of the data (e.g. seasonal variation) affects the observed spatial features. The spatio-temporal sampling of the collocated dataset is not even, but is affected e.g. by solar zenith angle, cloudiness, and snow cover. In particular, the Northern Hemisphere high latitude areas have a relatively strong seasonal cycle of XCO$_2$ (Lindqvist et al., 2015), which is not fully captured in this aggregated dataset, as the winter months are scarcely sampled (A3). We also emphasize that the OCO-2 swath is very narrow, and repeats over the same areas leaving relatively large gaps without data. The crude map presentation with $1° \times 1°$ lat/lon grid in Fig. 4 artificially fills the gaps and smooths the data, while the patchy structure of the data is still seen in the Northern high-latitude areas. Therefore, these maps serve only as rough reference indicating spatial variance in retrieved





XCO$_2$ values, and one should not draw far-reaching conclusions from it. More detailed analyses are made based on the statistics from the spatio-temporally collocated subsets of the full dataset in the following.

Figure 5 shows the retrieved XCO$_2$ values aggregated to the AOD matrix. When aggregating five years of data we first apply a simple linear trend correction in an attempt to remove the effect of increasing CO$_2$ values, as described in the Methods section. Figure 5 a) shows clearly, when considering all data points (no quality filtering), that the retrieved XCO$_2$ values are correlated with the relative AOD values. In AOD quarter Q4, where OCO-2 AOD is biased high compared to MODIS, we get lower XCO$_2$ values (1.3 ppm lower than the total average). In Q2, where OCO-2 AOD is biased low compared to MODIS, we

get higher XCO$_2$ values (0.4 ppm higher than the total average). When quality filtering is applied (Fig. 5 b) the total average is increased by 0.2 ppm, and the Q2 average is 0.5 ppm above the total average. Table 2 shows average XCO$_2$ values for quarters Q1 and Q2 for the quality filtered data (very few data remain in Q3 and Q4 after filtering). Table A4 summarises the average XCO$_2$ values in different AOD quarters for the unfiltered data.

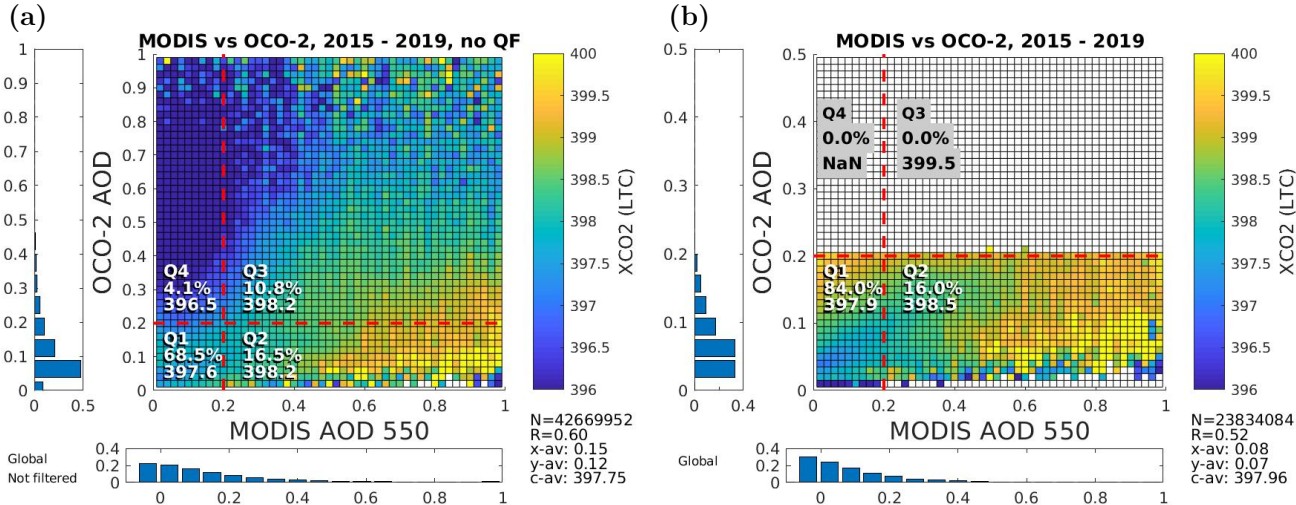

**Figure 5.** OCO-2 XCO$_2$ retrievals for five years aggregated to the AOD matrix. Linear trend correction (LTC) has been applied to the XCO$_2$ values. **a)** All data, **b)** only good quality data. The text insets on the scatter plot show the fraction of data in each AOD quarter and the mean XCO$_2$ value. The lower right hand text inset shows the number of data (N), correlation coefficient (R), average AOD values for MODIS (x-av) and OCO-2 (y-av), and average XCO$_2$ (c-av). The normalized histograms show the distribution of AOD data along each axes respectively.

    The striking connection between XCO$_2$ and the relative AOD values between the two instruments in Fig. 5 a) can potentially

be explained by the light path length used in the ACOS full physics retrieval. If the aerosol load is underestimated in the retrieval (Q2), the light path is also underestimated, and the measured CO$_2$ absorption is divided into too short distance, leading to overestimation of XCO$_2$. Similarly, if AOD is overestimated, the light path is also overestimated, causing underestimation of XCO$_2$. While the potentially bad-quality XCO$_2$ retrievals in Q3 and Q4 are removed in the quality filtering, the possible aerosol effects in Q2 remain in the quality filtered OCO-2 data.





| Dataset | $XCO_2$ (LTC) | | | $\Delta XCO_2$ | | | $XCO_2$ anom. | | | AOD | | |
|---|---|---|---|---|---|---|---|---|---|---|---|---|
| (Quality filtered) | Q1 | Q2 | Total | Q1 | Q2 | Total | Q1 | Q2 | Total | MODIS | OCO-2 | R |
| Global | 397.9 | 398.5 | 398.0 | -0.14 | 0.48 | -0.04 | -0.03 | 0.05 | -0.01 | 0.08 | 0.07 | 0.52 |
| Urban | 399.1 | 399.7 | 399.3 | 1.11 | 1.72 | 1.33 | 0.00 | 0.11 | 0.04 | 0.18 | 0.07 | 0.52 |
| TCCON(1) | 399.4 | 399.9 | 399.5 | 1.37 | 1.88 | 1.46 | -0.01 | 0.12 | 0.01 | 0.09 | 0.06 | 0.45 |
| TCCON(2) | 399.1 | 399.9 | 399.2 | 1.10 | 1.86 | 1.22 | | | | 0.09 | 0.06 | 0.45 |

**Table 2.** $XCO_2$ statistics for different good quality datasets for the two AOD quarters Q1 and Q2 (see text). $\Delta XCO_2$ is calculated with respect to the reference value 398.1 ppm (the total global average value). For collocated TCCON data two $XCO_2$ values are given, from OCO-2 (1) and from TCCON (2), respectively. The $XCO_2$ anomaly is calculated with respect to the OCO-2 median value within 500 km. MODIS AOD is calculated at 550 nm, OCO-2 total AOD at 755 nm; R is the correlation coefficient.

The correlation between $XCO_2$ and AOD can be a sign of a retrieval bias caused by aerosols, or it can be caused by real correlation between aerosols and $CO_2$ emissions. It is entirely plausible that there is a natural correlation between AOD and $XCO_2$, stemming partially from anthropogenic (or, in case of fires or volcanoes, natural) coemission of $CO_2$ and aerosols. However, the striking feature in Fig. 5 is the dependence of $XCO_2$ on the relative AOD values between the two instruments. This dependence of $XCO_2$ on the AOD difference implies that possible biases in the aerosol treatment have an effect on the

$XCO_2$ retrievals. In the following we will study these two possible causes of the observed correlation between $XCO_2$ and MODIS AOD in more detail. On one hand, to investigate the natural correlation, we will focus more on urban areas, where anthropogenic emissions are presumably more pronounced. On the other hand, we will consider the reference $XCO_2$ from 26 TCCON sites collocated with both OCO-2 and MODIS.

     We have created an urban subset of the collocated data using a MODIS-based urban areas mask (NaturalEarth, last access:

22 April 2024). Figure A7 shows the urban areas (and fraction of data in Q1 for these areas). The data is reduced to slightly below one million data points (2% of all data), with mean $XCO_2$ value 1.3 ppm higher than for the global data for the quality filtered case. Similarly to the global case, lower $XCO_2$ values in Q4 and higher values in Q2 are seen for the unfiltered data (Table A4). For the urban areas there is much higher fraction of high MODIS AOD data (Q2+Q3) than globally: 36.2% (45.7%) compared to 16.0% (27.3%) for filtered (unfiltered) data. Average MODIS AOD for urban areas is 0.18 (0.24), while for the

global data set it is 0.08 (0.15). Interestingly, for OCO-2 the corresponding values are 0.07 (0.11) and 0.07 (0.12), respectively and the high OCO-2 AOD fraction (Q3+Q4) is about the same for urban and global datasets (Table 2 and Table A4). It should be noted that earlier versions of MODIS DT aerosol retrieval had some issues over urban areas (Gupta et al., 2016), and more detailed studies on the reliability of the reference AOD values in urban areas might be useful.

     Finally, there is a column for $XCO_2$ anomaly in Table 2. The OCO-2 $XCO_2$ anomaly is calculated for each good quality

OCO-2 pixel in the collocated dataset as the difference from the median $XCO_2$ value calculated within 500 km for the corresponding OCO-2 orbit. This is an alternative way to de-seasonalize and de-trend the data, instead of applying the simple LTC. The idea is to study covariance of AOD values and local $XCO_2$ anomalies caused by possible $CO_2$ sources and sinks. We see that the average $XCO_2$ anomaly is negative (-0.03 ppm) in Q1 for the global dataset, indicating that average $XCO_2$ is lower in





low AOD areas. Also, the anomaly is higher in Q2 further supporting the idea that local $XCO_2$ positive anomaly (source) is

connected with higher AOD. For the urban areas the positive anomaly in Q2 is enhanced (0.11 ppm).

In order to further investigate to what extent the observed relation between AOD and $XCO_2$ are related to possible retrieval issues on the one hand and to the natural covariance of AOD and $XCO_2$ on the other hand, we have collocated the five year OCO-2/MODIS dataset with the ground-based data from 26 TCCON sites (see Table A6). From Table 2 we see that the TCCON $XCO_2$ is 0.8 ppm higher in Q2 than in Q1, suggesting that there is a real positive correlation between AOD and $XCO_2$. For the

OCO-2 $XCO_2$ values in the collocated TCCON dataset the difference between Q1 and Q2 is 0.5 ppm for the quality filtered data. The $XCO_2$ values are systematically higher in Q2 than in Q1 for all subsets, suggesting a positive correlation between MODIS AOD and OCO-2 $XCO_2$. In particular, the difference between Q2 and Q1 is highest for the TCCON $XCO_2$ data, which suggests that there is actually a stronger correlation between MODIS AOD and $XCO_2$ than suggested by the OCO-2 data.

Figure 6 shows joint histograms of $XCO_2$ and MODIS AOD with bivariate linear fits. In addition to the global dataset, the

urban and TCCON subsets are shown. There is a small but statistically significant correlation between $XCO_2$ and AOD, and this correlation is strongest when using the TCCON $XCO_2$ data. The linear fit also shows higher positive slope for TCCON. This suggests that there is a real correlation between AOD and $XCO_2$, and this correlation is partly masked by aerosol effects in the OCO-2 retrievals. Figure 7 shows combined bin-averaged plots and linear fits for the different subsets, also as function of OCO-2 AOD and AOD difference. For OCO-2 AOD the slopes are also positive (and steeper). Disentangling the the effects

of AOD difference between MODIS and OCO-2 and the dependence of $XCO_2$ and AOD makes interpretation of Fig. 7 c) complicated, but it is shown for completeness.

Figure A8 shows similar plots using also OCO-2 AOD and the AOD difference on the x-axis, and the $XCO_2$ difference between OCO-2 and TCCON on the y-axis, and reveals negative correlation coefficients and negative slope for the linear fits. Figure A8 a) shows a weak but statistically significant correlation between MODIS AOD and the OCO-2 $XCO_2$ bias with

respect to TCCON. OCO-2 seems to slightly overestimate $XCO_2$ for low AOD values, and underestimate at high AOD values.

A post-process correction, based on systematic comparisons with the TCCON data, is routinely applied to OCO-2 $XCO_2$ data (O'Dell et al., 2018). Even when using the bias-corrected data, our comparison with TCCON reveals a residual bias which depends on the MODIS AOD. These observations further support the suggested correlation between AOD and $XCO_2$, which is partly masked by aerosols effects in the satellite retrievals. Table 3 summarizes the observed correlation coefficients and linear

fit slopes for $XCO_2$ as function of AOD for the different datasets.

Disentangling the effects of AOD and $XCO_2$ differences in the comparison is not straightforward. One should also note that the TCCON sampling may affect the results. For example, not all of the included TCCON sites have data for the whole five year period. In particular, some sites with higher AOD and $XCO_2$ values are included only towards the end of the time period. We also see that the TCCON sites are not representative for the globe in the sense that the average OCO-2 $XCO_2$ value for the

TCCON subset is 1.5 ppm higher than the global average for data collocated with MODIS (Table 2). Most of the TCCON sites are located in the northern hemisphere, with large gaps between sites. A more detailed analysis considering individual TCCON sites respectively would be required to confirm the observed dependencies, and this is a subject of a separate study.





**Figure 6.** Dependence of $XCO_2$ (LTC) on MODIS AOD for different subsets (quality filtered data). The dashed red line shows binned mean $XCO_2$ values for MODIS AOD bins. The dashed green line shows corresponding bivariate linear fit. The box plot shows the interquartile range for an AOD bin, while the whiskers show 9th and 91st percentiles. The text inset on lower right corner shows similar information as in Fig. 2, with the addition of 95% confidence range for the correlation coefficient R in parentheses. **a)** Global collocated data set. **b)** Urban data subset. **c)** Collocated MODIS/OCO-2/TCCON dataset, showing OCO-2 $XCO_2$ values (with TCCON priori adjustment). **d)** Collocated MODIS/OCO-2/TCCON dataset, showing TCCON $XCO_2$ values (60 minute average centred at the OCO-2 overpass time).

## 4.4 Temporal and spatial dependence

The correlation between AOD and $XCO_2$ in the global collocated dataset for five years may be partly explained by spatial
and temporal (co)variance in AOD and $XCO_2$ (see Fig. 4 b) and Fig. A3 b). For example, seasonal (co)variability of AOD




| XCO$_2$ (LTC) | MODIS AOD 550 | | OCO-2 AOD | | AOD difference | |
|---|---|---|---|---|---|---|
| **Dataset** | R | Slope | R | Slope | R | Slope |
| Global | 0.10 | 1.80 | 0.16 | 10.46 | -0.06 | -1.24 |
| Urban | 0.16 | 2.30 | 0.04 | 2.38 | -0.17 | -2.66 |
| TCCON(1) | 0.12 | 2.15 | 0.18 | 15.33 | -0.09 | -1.72 |
| TCCON(2) | 0.17 | 2.86 | 0.25 | 19.89 | -0.13 | -2.32 |

**Table 3.** Correlation and bivariate linear regression slopes for XCO$_2$ vs AOD for different subsets, and for AOD from different instruments (p-values $< 10^{-6}$ for all cases). For the collocated TCCON dataset, XCO$_2$ values from OCO-2 (1) and TCCON (2) are used, respectively.

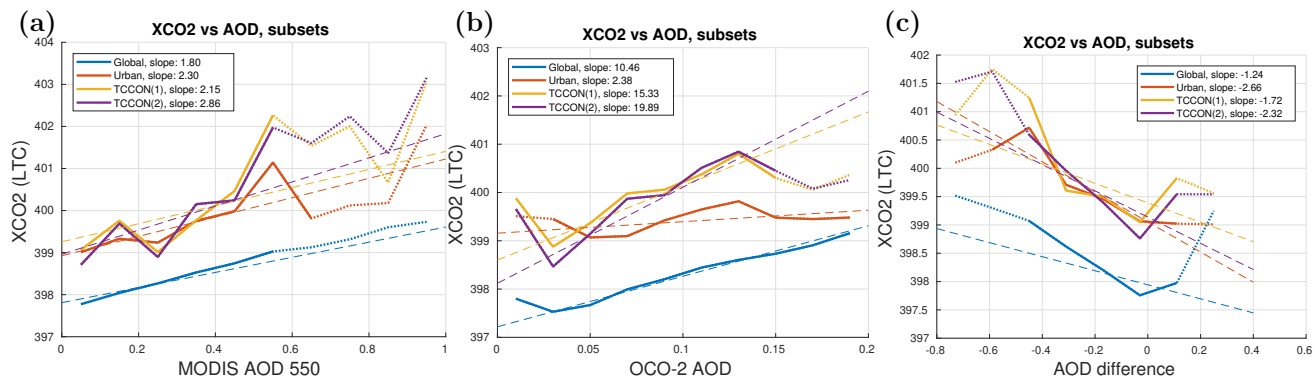

**Figure 7.** Dependence of bin-averaged XCO$_2$ (linear trend corrected, quality filtered) on AOD for three subsets of OCO-2 data and TCCON (solid lines). **a)** XCO$_2$ vs MODIS AOD. **b)** XCO$_2$ vs OCO-2 total AOD. **c)** XCO$_2$ vs AOD difference (OCO-2 - MODIS). The dashed lines show simple linear regression lines. The dotted parts of the bin-average lines correspond to AOD bins with less than 1% of all data.

and XCO$_2$ may affect the statistics, as well as areas with constant high/low aerosol load and CO$_2$ emission. In this section the spatial and temporal variability of the data are explored in some detail.

Figure 8 a) shows binned mean XCO$_2$ values for MODIS AODs bins and linear fits for different years, revealing a positive correlation between the two quantities for all years. The seasonal plot (Fig. 8 b) shows positive slopes for all seasons, with some seasonal variability with lower XCO$_2$ values in JJA and SON, and higher slope in MAM. We have further analysed the spatial distribution of data by studying respectively seven geographic areas: SE Asia, N Asia, N America, S America, Europe and Australia. The areas are defined in in Fig. A7. Figure 8 c) summarizes the results by showing the bin-averaged XCO$_2$ values for MODIS AOD bins. Most of the areas show a positive correlation between XCO$_2$ and AOD, with the clear exception of Northern Asia, which is dominated by strong seasonal cycle of XCO$_2$. We also note that the northern areas are strongly undersampled in winter months due to snow cover and high SZA, which prevent MODIS aerosol retrievals. Table A2 summarizes the statistics for the different subsets (global, urban, TCCON collocation, years, seasons, and areas). Table A3





shows further statistics per seasons for each geographic area. Table A5 summarises the correlations coefficients and slopes for different subsets.

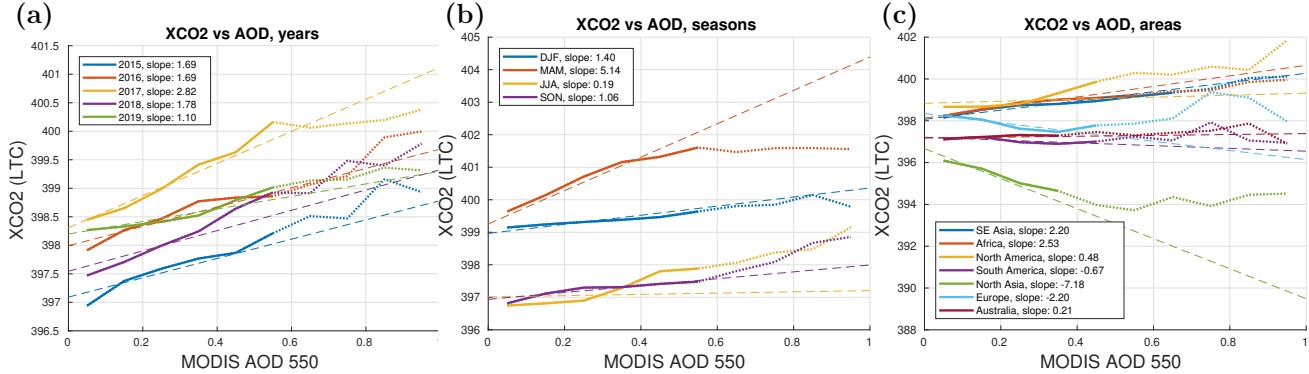

**Figure 8.** Dependence of bin-averaged OCO-2 $XCO_2$ (linear trend corrected, quality filtered) on MODIS AOD for **a)** different years, **b)** seasons (averaged over five years), and **c)** seven geographic areas (solid lines). The dashed lines show simple linear regression lines. The dotted parts of the bin-average lines correspond to AOD bins with less than 1% of all data. Note the different scale on y-axis.

## 4.5 Alternative AOD thresholds in anticipation of the CO2M

Satellite $XCO_2$ retrievals are known to have higher uncertainty in high aerosol conditions. Setting an AOD threshold for good quality retrievals is always a trade-off between coverage and quality of the data. While for OCO-2 a strict AOD threshold is used to ensure good quality retrievals, for the coming CO2M a good coverage over polluted regions is also crucial for monitoring $CO_2$ emissions. In the latter case it is also important to avoid possible sampling bias caused by excluding high AOD areas from analysis, considering the co-emission of anthropogenic aerosols and $CO_2$. In the upcoming CO2M mission the required AOD

threshold for good quality retrievals is designed to be 0.5 (ESA (Y. Meijer, Mission Scientist), last access: 23 April 2024).

Table 4 shows the fraction of collocated quality filtered data for two different MODIS AOD bins, using either 0.2 or 0.5 as the threshold for maximum AOD (at 550 nm). For the global dataset relaxing the MODIS AOD threshold from 0.2 to 0.5 increases the fraction of acceptable data by 14.4 percentage points, while the average $XCO_2$ is increased by 0.08 ppm. For the urban areas the increase in coverage is as high as 30.8 percentage points while the increase in $XCO_2$ is 0.14 ppm. This

finding support the idea that being able to perform reliable $XCO_2$ retrievals at higher aerosol loads is crucial for capturing the anthropogenic $CO_2$ emissions.

Figure 9 shows the fraction of data in the two considered MODIS AOD bins zoomed-in to South-East Asia which stands out as high AOD area. Here the two MODIS AOD bins are partly overlapping, M1: -0.2 - 0.2 and M2: -0.2 - 0.5. M1 contains 59% of the data, while M2 contains 94% of the data in this area. We see that large areas have a low fraction of data in M1, while for M2 only a few heavy AOD areas have low fraction of data within the bin. The high values over India and Eastern




China indicate that in these areas relaxing the MODIS AOD threshold from 0.2 to 0.5 increases the fraction of acceptable data considerably.

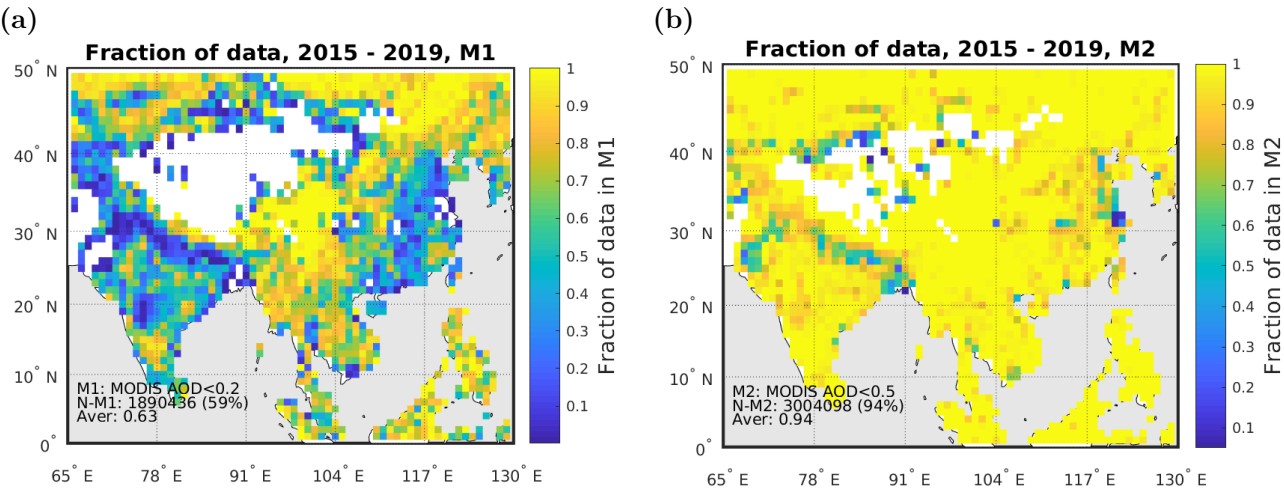

**Figure 9.** Difference in using MODIS AOD threshold 0.2 or 0.5 in Asia. Panels a) and b) show the fraction of data in M1 or M2, respectively, for each $1° \times 1°$ grid cell.

| Dataset | Fraction of data [%] | | | $XCO_2$ [ppm] | | |
|---|---|---|---|---|---|---|
| | AOD<0.2 | AOD<0.5 | $\Delta$ | AOD<0.2 | AOD<0.5 | $\Delta$ |
| Global | 84.0 | 98.3 | 14.4 | 397.9 | 397.9 | 0.08 |
| Urban | 63.8 | 94.6 | 30.8 | 399.1 | 399.3 | 0.14 |
| TCCON (1) | 83.5 | 98.1 | 14.6 | 399.4 | 399.4 | 0.03 |
| TCCON (2) | 83.5 | 98.1 | 14.6 | 399.1 | 399.2 | 0.07 |

**Table 4.** Difference between using 0.2 or 0.5 as MODIS AOD threshold (quality filtered data).

## 5   Conclusions

In this work, we have compiled and analysed a five-year dataset of co-located aerosol and $XCO_2$ satellite observations from
MODIS/Aqua and OCO-2, respectively. We have shown that the total AOD value in ACOS full physics retrieval differs considerably from the MODIS Dark Target AOD over land for a large fraction of the data. There is also co-variability between the AOD difference and the retrieved $XCO_2$ values, most strikingly visible for the unfiltered dataset: overestimation of AOD in the OCO-2 retrieval leads to lower $XCO_2$ values, and underestimation of AOD leads to higher $XCO_2$ values. We have also





found evidence of real covariance of AOD and $XCO_2$, which is partly masked by the aerosol effect on the $XCO_2$ retrieval. This
covariance is presumably at least in part due to co-emission of anthropogenic $CO_2$ and aerosols.

Heavy aerosol conditions are known to hamper the satellite $XCO_2$ retrievals such that to ensure the quality of the retrievals
an upper AOD threshold needs to be applied. Setting the threshold is always a trade-off between coverage and quality of the
retrievals, and removing retrievals over areas of constant elevated aerosol loads, such as many urban areas, will unavoidably lead
to a sampling bias in the quality filtered data. With the upcoming CO2M mission, where the goal is to contribute observations
of anthropogenic $CO_2$ to the Global Stocktake, this necessity of a larger AOD threshold is a major challenge. Here we have
studied the effect of different AOD thresholds on the coverage of retrieved $XCO_2$ values.

First, we note that the current quality filtering in the ACOS retrieval, which effectively removes most datapoints with AOD
larger than 0.2, works relatively well in removing $XCO_2$ retrievals where AOD is overestimated compared to MODIS. However,
a large fraction of data remains where the AOD estimated in ACOS retrieval is lower than in the collocated MODIS retrievals.
We have tested relaxing the MODIS AOD threshold from 0.2 to 0.5 for the quality-filtered collocated OCO-2/MODIS dataset,
and found that the increase in accepted datapoints is 14.4 percentage points globally and 30.8 percentage points for the urban
areas. In anticipation of the CO2M this supports the notion that an AOD threshold as high as 0.5 would be crucial to capture
the $CO_2$ emissions in urban areas and to attain global observational coverage mainly by a significantly improved coverage over
Asia.

We note that even though we have not attempted to improve the aerosol treatment of the ACOS retrieval in this work,
the collocated MODIS/OCO-2 dataset could be used in post-processing to remove quality-filtered data points that have high
MODIS AOD, indicating possible aerosol effects in the $XCO_2$ retrievals. Such filtering could be used, for example, before
assimilating the OCO-2 data into models.

Finally, the focus in this paper has been on the global multiyear statistics of AOD and $XCO_2$ in the collocated satellite
dataset. The comparison with ground-based TCCON data has been done only statistically, combining all sites, to give a first
reference point to independent data. A more detailed study focusing carefully on the sampling and representativeness of data
and the specific conditions of each TCCON site would be needed to make informed conclusions on the retrieval performance
in heavy aerosol conditions.

*Code and data availability.*

The AERONET data is available from NASA Goddard Space Flight Center at https://aeronet.gsfc.nasa.gov/new_web/data.html.
The TCCON data were obtained from the TCCON Data Archive hosted by CaltechDATA at https://tccondata.org (see Table A6
for references). The MODIS data used in this work can be found and downloaded using the NASA Earthdata Search website
at https://www.earthdata.nasa.gov/. The OCO-2 data were produced by the OCO-2 project at the Jet Propulsion Laboratory,
California Institute of Technology, and obtained from the OCO-2 data archive maintained at the NASA Goddard Earth Science
Data and Information Services Center (OCO-2 Science Team et al., 2020). The collocated OCO-2/MODIS dataset created in
this work and related codes are available as open data (Virtanen, 2024).





# Appendix A: Supplementary data

## A1 Supplementary Tables

| Year | | | | All data (unfiltered) | | | | Good quality (Filtered) | | | |
|---|---|---|---|---|---|---|---|---|---|---|---|
| | OCO2 | Fraction | | AOD | | | XCO$_2$ | AOD | | | XCO$_2$ |
| | N [$10^6$] | MOD | QF | MOD | OCO2 | R | [ppm] | MOD | OCO2 | R | [ppm] |
| 2015 | 52.2M | 16.3 | 56.2 | 0.15 | 0.12 | 0.54 | 398.3 | 0.08 | 0.07 | 0.54 | 398.5 |
| 2016 | 67.1M | 13.6 | 57.2 | 0.15 | 0.11 | 0.64 | 401.6 | 0.08 | 0.07 | 0.53 | 401.8 |
| 2017 | 55.4M | 13.1 | 57.7 | 0.14 | 0.12 | 0.62 | 404.4 | 0.08 | 0.07 | 0.52 | 404.5 |
| 2018 | 66.9M | 13.6 | 56.4 | 0.15 | 0.12 | 0.63 | 406.0 | 0.09 | 0.07 | 0.52 | 406.2 |
| 2019 | 66.4M | 13.1 | 52.0 | 0.15 | 0.12 | 0.61 | 408.8 | 0.08 | 0.07 | 0.50 | 409.2 |
| Total | 308.0M | 13.9 | 55.9 | 0.15 | 0.12 | 0.60 | 403.8 | 0.08 | 0.07 | 0.52 | 404.0 |

**Table A1.** Number of data and average AOD and XCO$_2$ values for the five years of collocated OCO-2 and MODIS DT-land data considered in this work. Second column ('OCO2') shows the number of original OCO-2 data (in millions) for each year. The next column ('MOD') shows the fraction of OCO-2 data which have a matching MODIS AOD observation. The fourth column ('QF') shows the fraction of collocated data after OCO-2 quality filter has been applied (with respect to all collocated data). Also shown are the yearly average OCO-2 XCO$_2$ value and AOD value for each instrument, and the correlation coefficient (R) between the collocated AOD data for the unfiltered and filtered data, respectively.

## A2 Supplementary Figures

*Author contributions.*

TV, HL and AS conceptualized the study. AL did initial MODIS data processing. TV did most of the data processing and visualization. HL did the OCO-2/TCCON data collocation and OCO-2 anomaly data calculations. RN did data and image processing for the collocated AERONET/OCO-2 data. First draft of the manuscript was written by TV and HL. All authors contributed to editing the final version of the manuscript.

*Competing interests.*

The authors declare that they have no conflict of interest.

**(a)**

**Gridded Match fraction, 2018**

**(b)**

**MODIS/OCO-2 dataset, Fraction of good quality data, 2015 - 2019**

**Figure A1. a)** Fraction of OCO-2 datapoints (without quality filtering) with matching MODIS data for $1° \times 1°$ grid cells. The fraction is only shown for grid cells which have at least one MODIS data point. $N$ means total number of MODIS data points. The average fraction (64%) of OCO-2 data points with matching MODIS data is calculated over those grid cells, which have non-zero fraction. Example for one year, 2018. **b)** Fraction of good quality data for each $1° \times 1°$ grid cell in the collocated MODIS/OCO-2 dataset.

*Acknowledgements.* We thank the AERONET PIs and their staff for establishing and maintaining the sites used in this investigation. We thank the PIs and others operating the TCCON sites for providing the data (Table A6). We thank the MODIS and OCO-2 science teams for the data used in this study. This work was supported by the Research Council of Finland (projects 331829, 337552, 353082, and 359196).



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





| Filtered Dataset | Fraction | | XCO$_2$ (LTC) | | XCO$_2$ anomaly | | AOD | | | |
|---|---|---|---|---|---|---|---|---|---|---|
| | Q1 | Q2 | Q1 | Q2 | Q1 | Q2 | MOD | OCO2 | R | N |
| Global | 84.0 | 16.0 | 397.9 | 398.5 | -0.03 | 0.05 | 0.08 | 0.07 | 0.52 | 23.8M |
| Urban | 63.8 | 36.2 | 399.1 | 399.7 | 0.00 | 0.11 | 0.18 | 0.07 | 0.52 | 535k |
| TCCON(1) | 83.5 | 16.5 | 399.4 | 399.9 | -0.01 | 0.12 | 0.09 | 0.06 | 0.45 | 680k |
| TCCON(2) | 83.5 | 16.5 | 399.1 | 399.9 | | | 0.09 | 0.06 | 0.45 | 680k |
| 2015 | 84.5 | 15.5 | 397.1 | 397.7 | -0.02 | 0.05 | 0.08 | 0.07 | 0.54 | 4.8M |
| 2016 | 84.3 | 15.7 | 398.0 | 398.7 | -0.02 | 0.06 | 0.08 | 0.07 | 0.53 | 5.2M |
| 2017 | 84.0 | 16.0 | 398.4 | 399.3 | -0.03 | 0.05 | 0.08 | 0.07 | 0.52 | 4.2M |
| 2018 | 83.6 | 16.4 | 397.6 | 398.3 | -0.03 | 0.03 | 0.09 | 0.07 | 0.52 | 5.1M |
| 2019 | 83.2 | 16.8 | 398.2 | 398.6 | -0.03 | 0.06 | 0.08 | 0.07 | 0.50 | 4.5M |
| DJF | 79.7 | 20.3 | 399.0 | 399.4 | -0.01 | 0.10 | 0.10 | 0.08 | 0.44 | 4.3M |
| MAM | 85.5 | 14.5 | 399.4 | 401.0 | -0.04 | 0.02 | 0.08 | 0.07 | 0.50 | 5.1M |
| JJA | 84.9 | 15.1 | 397.0 | 397.2 | -0.04 | -0.02 | 0.08 | 0.07 | 0.59 | 8.5M |
| SON | 84.4 | 15.6 | 397.0 | 397.4 | -0.01 | 0.13 | 0.08 | 0.07 | 0.52 | 5.9M |
| SE Asia | 59.4 | 40.6 | 398.2 | 398.9 | -0.03 | 0.09 | 0.19 | 0.10 | 0.54 | 3.2M |
| Africa | 83.6 | 16.4 | 398.2 | 399.0 | 0.01 | 0.15 | 0.08 | 0.08 | 0.50 | 5.2M |
| N America | 87.6 | 12.4 | 398.8 | 399.2 | -0.02 | 0.03 | 0.07 | 0.06 | 0.38 | 4.5M |
| S America | 90.6 | 9.4 | 397.2 | 397.0 | -0.01 | 0.03 | 0.05 | 0.06 | 0.35 | 2.1M |
| N Asia | 93.0 | 7.0 | 396.4 | 394.8 | -0.04 | -0.22 | 0.05 | 0.07 | 0.51 | 2.2M |
| Europe | 87.4 | 12.6 | 398.3 | 397.6 | -0.09 | -0.14 | 0.08 | 0.07 | 0.55 | 2.2M |
| Australia | 97.5 | 2.5 | 397.2 | 397.3 | -0.01 | 0.12 | -0.00 | 0.05 | 0.22 | 2.4M |
| **Urban** | | | | | | | | | | |
| SE Asia | 31.0 | 69.0 | 399.1 | 399.6 | 0.08 | 0.22 | 0.33 | 0.11 | 0.52 | 59k |
| Africa | 65.3 | 34.7 | 398.4 | 399.2 | 0.06 | 0.10 | 0.16 | 0.08 | 0.58 | 31k |
| N America | 63.4 | 36.6 | 400.0 | 400.6 | 0.01 | 0.25 | 0.18 | 0.05 | 0.33 | 144k |
| S America | 83.3 | 16.7 | 396.9 | 396.8 | -0.06 | 0.10 | 0.09 | 0.07 | 0.28 | 11k |
| N Asia | 93.2 | 6.8 | 397.0 | 395.7 | 0.04 | -0.14 | 0.06 | 0.07 | 0.50 | 6k |
| Europe | 82.3 | 17.7 | 399.0 | 398.0 | -0.02 | -0.11 | 0.10 | 0.07 | 0.58 | 140k |
| Australia | 95.4 | 4.6 | 397.4 | 397.7 | -0.00 | 0.06 | 0.04 | 0.05 | 0.21 | 30k |

**Table A2.** Statistics for for different subsets of the collocated good quality OCO-2 vs MODIS dataset. Fraction of data and average XCO$_2$ values for two AOD quarters are given: OCO-2 AOD is smaller than 0.2 for both Q1 and Q2, while for MODIS AOD is smaller than 0.2 in Q1 and larger in Q2. For each case the correlation coefficient (R) between the AOD values from each instrument (p-values $< 10^{-6}$ for all cases), and number of data (N) are also given. The XCO$_2$ anomaly is calculated with respect to the OCO-2 median value within 500 km on each orbit (before aggregation; see 'Methods'). For collocated TCCON data two XCO$_2$ values are given, from OCO-2 (1) and from TCCON (2), respectively. Seasonal data (DJF, MAM, JJA, SON) are collected from the five years. Geographic regions are defined in Supplementary Fig. A5.





| Area/ | Fraction | | XCO$_2$ (LTC) | | XCO$_2$ anom. | | AOD | | | | Seas. |
|---|---|---|---|---|---|---|---|---|---|---|---|
| Season | Q1 | Q2 | Q1 | Q2 | Q1 | Q2 | MOD | OCO2 | R | N | frac. |
| **SE Asia** | | | | | | | | | | | |
| DJF | 54.2 | 45.8 | 399.4 | 399.5 | -0.07 | 0.05 | 0.21 | 0.09 | 0.63 | 1.0M | 32.7 |
| MAM | 59.3 | 40.7 | 401.2 | 401.6 | -0.04 | 0.08 | 0.18 | 0.12 | 0.42 | 623k | 19.5 |
| JJA | 64.0 | 36.0 | 395.5 | 396.5 | 0.02 | 0.10 | 0.17 | 0.10 | 0.48 | 559k | 17.5 |
| SON | 62.3 | 37.7 | 396.9 | 397.5 | -0.01 | 0.14 | 0.18 | 0.09 | 0.65 | 964k | 30.3 |
| **Africa** | | | | | | | | | | | |
| DJF | 83.8 | 16.2 | 399.2 | 400.2 | 0.05 | 0.24 | 0.08 | 0.10 | 0.30 | 1.2M | 23.5 |
| MAM | 89.5 | 10.5 | 397.4 | 399.8 | -0.02 | 0.02 | 0.05 | 0.06 | 0.55 | 1.1M | 20.3 |
| JJA | 82.3 | 17.7 | 398.2 | 398.7 | -0.02 | 0.13 | 0.08 | 0.07 | 0.68 | 1.9M | 36.5 |
| SON | 79.9 | 20.1 | 397.8 | 397.9 | 0.03 | 0.16 | 0.09 | 0.10 | 0.43 | 1.0M | 19.7 |
| **N America** | | | | | | | | | | | |
| DJF | 94.2 | 5.8 | 400.2 | 400.4 | -0.01 | 0.16 | 0.04 | 0.04 | 0.29 | 855k | 19.0 |
| MAM | 83.5 | 16.5 | 401.6 | 401.6 | -0.04 | 0.01 | 0.09 | 0.07 | 0.22 | 993k | 22.0 |
| JJA | 80.8 | 19.2 | 397.6 | 398.2 | -0.04 | -0.01 | 0.11 | 0.07 | 0.31 | 1.2M | 26.9 |
| SON | 92.2 | 7.8 | 397.2 | 397.2 | -0.01 | 0.11 | 0.05 | 0.04 | 0.40 | 1.4M | 32.1 |
| **S America** | | | | | | | | | | | |
| DJF | 81.9 | 18.1 | 396.6 | 396.4 | -0.03 | 0.04 | 0.09 | 0.07 | 0.09 | 541k | 25.4 |
| MAM | 93.9 | 6.1 | 396.9 | 397.4 | -0.00 | -0.03 | 0.05 | 0.05 | 0.29 | 417k | 19.6 |
| JJA | 96.0 | 4.0 | 398.0 | 398.5 | 0.00 | 0.01 | 0.02 | 0.05 | 0.57 | 683k | 32.0 |
| SON | 89.7 | 10.3 | 397.0 | 397.1 | -0.02 | 0.04 | 0.05 | 0.07 | 0.34 | 491k | 23.0 |
| **N Asia** | | | | | | | | | | | |
| DJF | 100.0 | 0.0 | 400.2 | NaN | -0.10 | NaN | 0.15 | 0.04 | -0.39 | 212 | 0.0 |
| MAM | 95.5 | 4.5 | 401.2 | 401.6 | -0.00 | 0.09 | 0.03 | 0.08 | 0.28 | 400k | 17.8 |
| JJA | 91.5 | 8.5 | 395.4 | 393.8 | -0.06 | -0.28 | 0.06 | 0.07 | 0.59 | 1.4M | 64.4 |
| SON | 96.2 | 3.8 | 395.3 | 395.3 | -0.00 | -0.09 | 0.04 | 0.05 | 0.55 | 399k | 17.8 |
| **Europe** | | | | | | | | | | | |
| DJF | 95.9 | 4.1 | 400.7 | 401.1 | -0.02 | 0.09 | 0.05 | 0.04 | 0.50 | 206k | 9.2 |
| MAM | 89.6 | 10.4 | 401.1 | 400.9 | -0.16 | -0.16 | 0.07 | 0.07 | 0.50 | 593k | 26.5 |
| JJA | 81.6 | 18.4 | 396.7 | 396.5 | -0.09 | -0.22 | 0.10 | 0.08 | 0.56 | 881k | 39.4 |
| SON | 91.4 | 8.6 | 396.5 | 396.6 | -0.07 | 0.12 | 0.07 | 0.06 | 0.53 | 555k | 24.8 |
| **Australia** | | | | | | | | | | | |
| DJF | 95.1 | 4.9 | 397.0 | 397.3 | -0.03 | 0.06 | 0.02 | 0.07 | 0.16 | 296k | 12.2 |
| MAM | 98.2 | 1.8 | 396.7 | 396.8 | -0.02 | 0.16 | -0.01 | 0.05 | 0.20 | 587k | 24.2 |
| JJA | 98.8 | 1.2 | 397.5 | 397.6 | -0.01 | 0.08 | -0.02 | 0.04 | 0.16 | 965k | 39.8 |
| SON | 95.9 | 4.1 | 397.2 | 397.5 | 0.01 | 0.15 | 0.01 | 0.07 | 0.18 | 574k | 23.7 |

**Table A3.** Same as Supplementary Table 2 above, but with seasonal statistics for seven areas, respectively (quality filtered data; p-values $< 10^{-6}$ for all cases). Also, the fraction of data (in %) for seasons is given for each area, respectively.





| Not filtered | **Fraction of data** | | | | $\Delta$**XCO$_2$** (ref: 398.1) [ppm] | | | | **AOD** | | | |
|---|---|---|---|---|---|---|---|---|---|---|---|---|
| **Dataset** | **Q1** | **Q2** | **Q3** | **Q4** | **Q1** | **Q2** | **Q3** | **Q4** | **MOD** | **OCO-2** | **R** | **N** |
| Global | 68.5 | 16.5 | 10.8 | 4.1 | -0.37 | 0.25 | 0.22 | -1.46 | 0.15 | 0.12 | 0.60 | 42.7M |
| Urban | 52.9 | 34.2 | 11.5 | 1.5 | 0.88 | 1.70 | 0.77 | -2.74 | 0.24 | 0.11 | 0.63 | 876k |
| TCCON(1) | 77.0 | 17.9 | 3.2 | 1.9 | 1.11 | 1.52 | 1.85 | -1.18 | 0.12 | 0.08 | 0.51 | 1.0M |
| TCCON(2) | 77.0 | 17.9 | 3.2 | 1.9 | 1.02 | 1.88 | 2.55 | 0.77 | 0.12 | 0.08 | 0.51 | 1.0M |
| 2015 | 68.7 | 16.3 | 10.9 | 4.1 | -1.09 | -0.53 | -0.87 | -2.64 | 0.15 | 0.12 | 0.54 | 8.5M |
| 2016 | 69.4 | 16.6 | 10.4 | 3.7 | -0.18 | 0.51 | 0.47 | -1.36 | 0.15 | 0.11 | 0.64 | 9.1M |
| 2017 | 69.4 | 16.2 | 10.2 | 4.3 | 0.18 | 1.16 | 1.39 | -0.76 | 0.14 | 0.12 | 0.62 | 7.3M |
| 2018 | 68.1 | 16.6 | 11.1 | 4.1 | -0.62 | 0.03 | 0.13 | -1.57 | 0.15 | 0.12 | 0.63 | 9.1M |
| 2019 | 67.2 | 16.8 | 11.6 | 4.4 | -0.09 | 0.20 | 0.23 | -0.91 | 0.15 | 0.12 | 0.61 | 8.7M |
| DJF | 64.4 | 19.9 | 11.0 | 4.6 | 0.70 | 1.30 | 1.87 | -0.57 | 0.17 | 0.13 | 0.52 | 7.9M |
| MAM | 66.5 | 15.1 | 13.0 | 5.4 | 1.16 | 2.71 | 3.08 | 0.70 | 0.15 | 0.13 | 0.60 | 9.6M |
| JJA | 70.7 | 15.9 | 10.0 | 3.4 | -1.23 | -1.18 | -1.92 | -3.37 | 0.14 | 0.11 | 0.68 | 14.6M |
| SON | 70.5 | 16.1 | 9.8 | 3.6 | -1.26 | -0.89 | -1.62 | -2.78 | 0.14 | 0.11 | 0.62 | 10.5M |
| SE Asia | 38.4 | 31.5 | 26.8 | 3.3 | 0.14 | 0.82 | 0.93 | 0.04 | 0.32 | 0.17 | 0.63 | 7.5M |
| Africa | 61.0 | 16.0 | 15.4 | 7.6 | 0.03 | 0.79 | 1.10 | -0.32 | 0.16 | 0.15 | 0.63 | 9.5M |
| N America | 79.2 | 13.8 | 4.3 | 2.7 | 0.66 | 0.81 | -1.81 | -2.21 | 0.10 | 0.08 | 0.53 | 6.7M |
| S America | 80.6 | 9.8 | 4.7 | 4.9 | -0.92 | -0.93 | -1.21 | -3.03 | 0.08 | 0.10 | 0.36 | 5.2M |
| N Asia | 83.6 | 9.4 | 4.5 | 2.5 | -1.86 | -3.32 | -5.23 | -3.65 | 0.10 | 0.09 | 0.71 | 3.7M |
| Europe | 77.4 | 15.0 | 5.1 | 2.5 | 0.00 | -0.64 | -2.32 | -2.67 | 0.11 | 0.09 | 0.58 | 3.6M |
| Australia | 93.3 | 3.3 | 0.7 | 2.7 | -1.08 | -0.98 | -2.46 | -4.33 | 0.01 | 0.07 | 0.31 | 3.1M |

**Table A4.** XCO$_2$ statistics in different datasets without OCO-2 quality filtering. For TCCON collocation XCO$_2$ is obtained from OCO-2 (1) and TCCON (2). Anomaly data is not available for the unfiltered case, instead $\Delta$XCO$_2$ is calculated with respect to the reference value 398.1 ppm (the total global average value for good quality data). Linear trend correction (LTC) has been applied. p-values $< 10^{-6}$ for all cases.





| Dataset | XCO$_2$ vs MODIS AOD | | XCO$_2$ vs OCO-2 AOD | | XCO$_2$ vs AOD difference | | AOD vs AOD | |
|---|---|---|---|---|---|---|---|---|
| | R | Slope | R | Slope | R | Slope | R | Slope |
| Global | 0.10 | 1.80 | 0.16 | 10.46 | -0.06 | -1.24 | 0.52 | 0.18 |
| Urban | 0.16 | 2.30 | 0.04 | 2.38 | -0.17 | -2.66 | 0.52 | 0.12 |
| TCCON(1) | 0.12 | 2.15 | 0.18 | 15.33 | -0.09 | -1.72 | 0.45 | 0.12 |
| TCCON(2) | 0.17 | 2.86 | 0.25 | 19.89 | -0.13 | -2.32 | 0.45 | 0.12 |
| 2015 | 0.09 | 1.69 | 0.12 | 8.17 | -0.06 | -1.38 | 0.54 | 0.18 |
| 2016 | 0.09 | 1.69 | 0.19 | 12.45 | -0.05 | -0.98 | 0.53 | 0.18 |
| 2017 | 0.15 | 2.82 | 0.22 | 13.95 | -0.10 | -2.14 | 0.52 | 0.19 |
| 2018 | 0.10 | 1.78 | 0.19 | 12.51 | -0.05 | -0.96 | 0.52 | 0.19 |
| 2019 | 0.07 | 1.10 | 0.10 | 5.94 | -0.04 | -0.80 | 0.50 | 0.18 |
| DJF | 0.10 | 1.40 | 0.11 | 5.24 | -0.08 | -1.17 | 0.44 | 0.17 |
| MAM | 0.27 | 5.14 | 0.42 | 27.88 | -0.17 | -3.60 | 0.50 | 0.18 |
| JJA | 0.01 | 0.19 | 0.00* | 0.07 | -0.01 | -0.25 | 0.59 | 0.19 |
| SON | 0.08 | 1.06 | 0.10 | 4.48 | -0.06 | -0.87 | 0.52 | 0.19 |
| SE Asia | 0.15 | 2.20 | 0.23 | 14.66 | -0.10 | -1.75 | 0.54 | 0.13 |
| Africa | 0.20 | 2.53 | 0.34 | 13.15 | -0.10 | -1.47 | 0.50 | 0.21 |
| N America | 0.02 | 0.48 | 0.09 | 7.78 | 0.00* | 0.05 | 0.38 | 0.13 |
| S America | -0.04 | -0.67 | 0.12 | 6.06 | 0.08 | 1.33 | 0.35 | 0.12 |
| N Asia | -0.20 | -7.18 | 0.06 | 6.71 | 0.25 | 10.28 | 0.51 | 0.20 |
| Europe | -0.08 | -2.20 | -0.07 | -6.11 | 0.06 | 2.06 | 0.55 | 0.21 |
| Australia | 0.02 | 0.21 | -0.01 | -0.38 | -0.02 | -0.26 | 0.22 | 0.09 |

**Table A5.** Statistics for correlation between AOD and XCO$_2$ and bivariate linear regression slopes for different subsets of the quality filtered collocated MODIS/OCO-2 five year (2015-2019) dataset. The first three slopes (columns 3, 5, and 7) are for XCO$_2$ as function of AOD (or AOD difference), while the last column gives the fitted slope for OCO-2 AOD as function of MODIS AOD. p-values are smaller than $10^{-6}$ except for the cases marked with $^*$; for these cases p$< 10^{-4}$.



| Site Name | Location | Data Citation |
|---|---|---|
| bremen01 | Bremen, Germany | Notholt et al. (2022) |
| burgos01 | Burgos, Philippines | Morino et al. (2022c) |
| pasadena01 | Pasadena, California, USA | Wennberg et al. (2022c) |
| easttroutlake01 | East Trout Lake, Canada | Wunch et al. (2022) |
| edwards01 | AFRC, Edwards, CA, USA | Iraci et al. (2022b) |
| eureka01 | Eureka, Canada | Strong et al. (2022) |
| garmisch01 | Garmisch, Germany | Sussmann and Rettinger (2017) |
| indianapolis01 | Indianapolis, Indiana, USA | Iraci et al. (2022a) |
| izana01 | Izana, Tenerife, Spain | Blumenstock et al. (2017) |
| jpl02 | JPL, Pasadena, California, USA | Wennberg et al. (2022a) |
| saga01 | Saga, Japan | Shiomi et al. (2022) |
| karlsruhe01 | Karlsruhe, Germany | Hase et al. (2022) |
| lauder02 | Lauder, New Zealand | Sherlock et al. (2022) |
| lauder03 | Lauder, New Zealand | Pollard et al. (2022) |
| manaus01 | Manaus, Brazil | Dubey et al. (2022) |
| nicosia01 | Nicosia, Cyprus | Petri et al. (2023) |
| nyalesund01 | Ny-Ålesund, Svalbard, Norway | Buschmann et al. (2022) |
| lamont01 | Lamont, Oklahoma, USA | Wennberg et al. (2022d) |
| orleans01 | Orleans, France | Warneke et al. (2022) |
| parkfalls01 | Park Falls, Wisconsin, USA | Wennberg et al. (2022b) |
| paris01 | Sorbonne Université, Paris, FR | Te et al. (2022) |
| reunion01 | Reunion Island, France | Maziere et al. (2022) |
| rikubetsu01 | Rikubetsu, Hokkaido, Japan | Morino et al. (2022a) |
| sodankyla01 | Sodankylä, Finland | Kivi et al. (2022) |
| tsukuba02 | Tsukuba, Ibaraki, Japan, 125HR | Morino et al. (2022b) |
| xianghe01 | Xianghe, China | Zhou et al. (2022) |

**Table A6.** 26 TCCON sites used in this study.



(a)


(b)

(c)

**Figure A2.** Same as Fig. 1 in the paper, but for MODIS Deep Blue (DB).



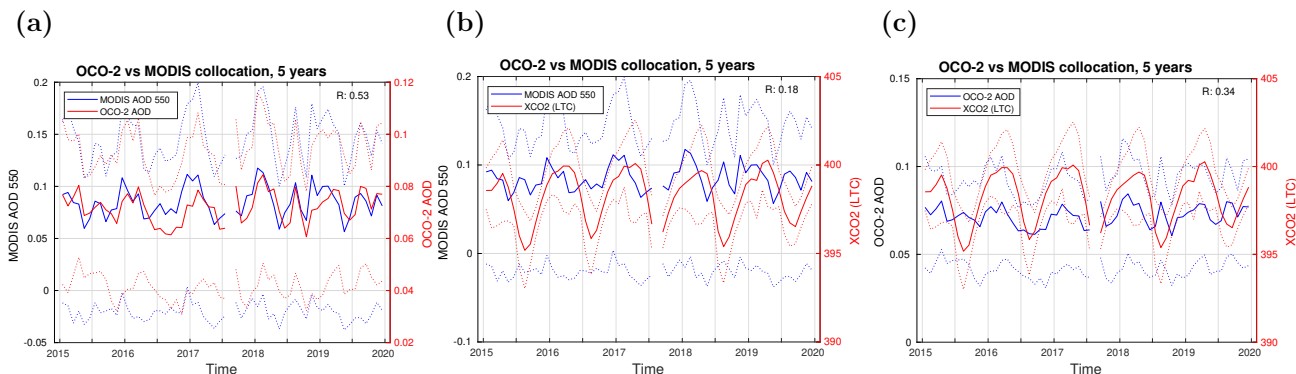

**Figure A3.** Temporal bin plots (3-week mean values) for the global, quality filtered collocated OCO-2/MODIS dataset. Dotted lines show the interquartile range. Correlation coefficients R are calculated from the temporal bin values. Comparison of **a)** MODIS and OCO-2 AOD, **b)** MODIS AOD and OCO-2 $XCO_2$, **c)** OCO-2 AOD and XCO-2. The positive correlation suggests that there is temporal covariance between AOD and $XCO_2$.





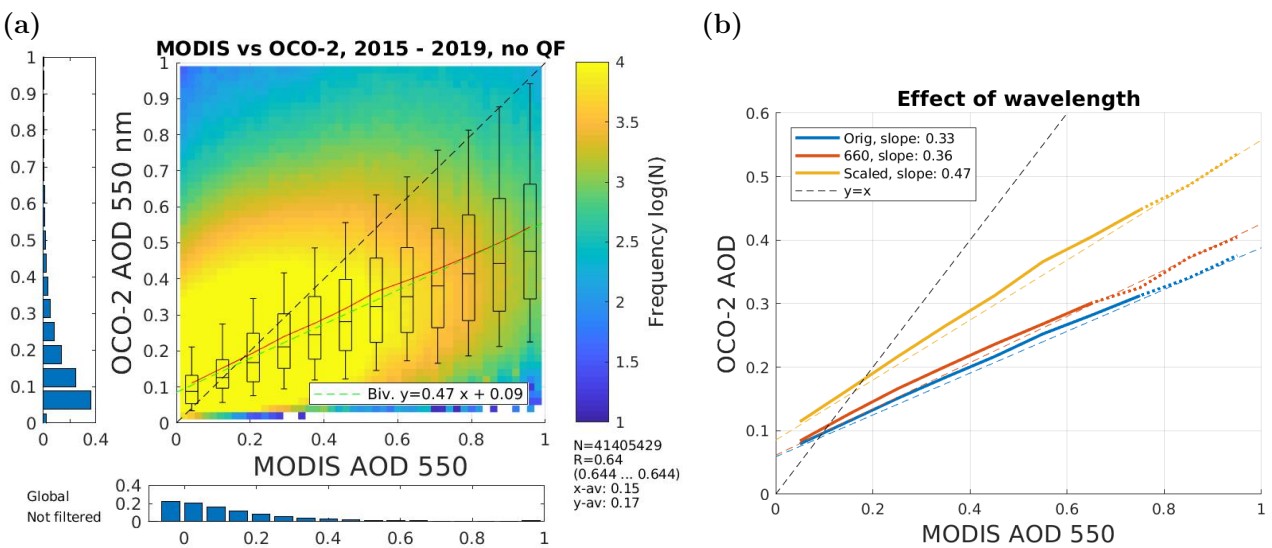

**Figure A4. a)** AOD comparison with OCO-2 AOD scaled from 755 nm to 550 nm using the Angstrom exponent from collocated MERRA-2 data. The red line shows binned mean values, the dashed green line shows bivariate linear fit, the boxes show interquartile range and the whiskers show 9th and 91st percentiles for MODIS AOD bins. **b)** Comparison of OCO-2 AOD against MODIS AOD for three different cases: the original comparison between MODIS AOD at 550 nm and OCO-2 total AOD at 755 nm ('Orig', blue line), MODIS AOD at 660 nm vs OCO-2 AOD at 755 nm ('660', red line); MODIS AOD at 550 nm vs OCO-2 AOD scaled to 550 nm ('scaled', yellow line). Solid lines show bin-averaged OCO-2 AOD (for MODIS AOD bins); the dotted part correspond to bins with less than 1% of all data. Dashed lines show bivariate linear fits. OCO-2 quality filtering has not been applied.

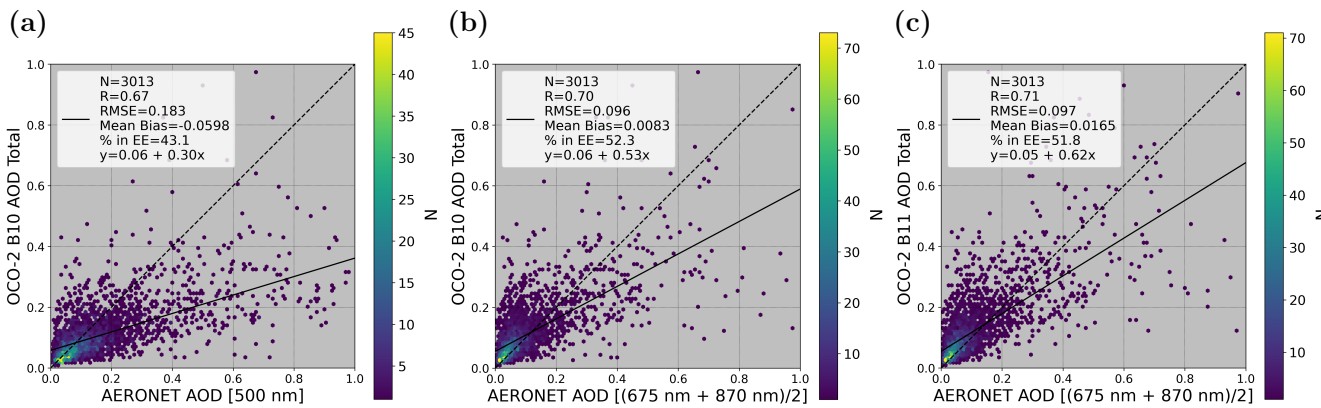

**Figure A5.** Comparison between OCO-2 and AERONET for all collocated data through February 2023. **a)** AERONET AOD at 500nm. **b)** AERONET AOD scaled to 770 nm by simple average. **c)** OCO-2 version B11.





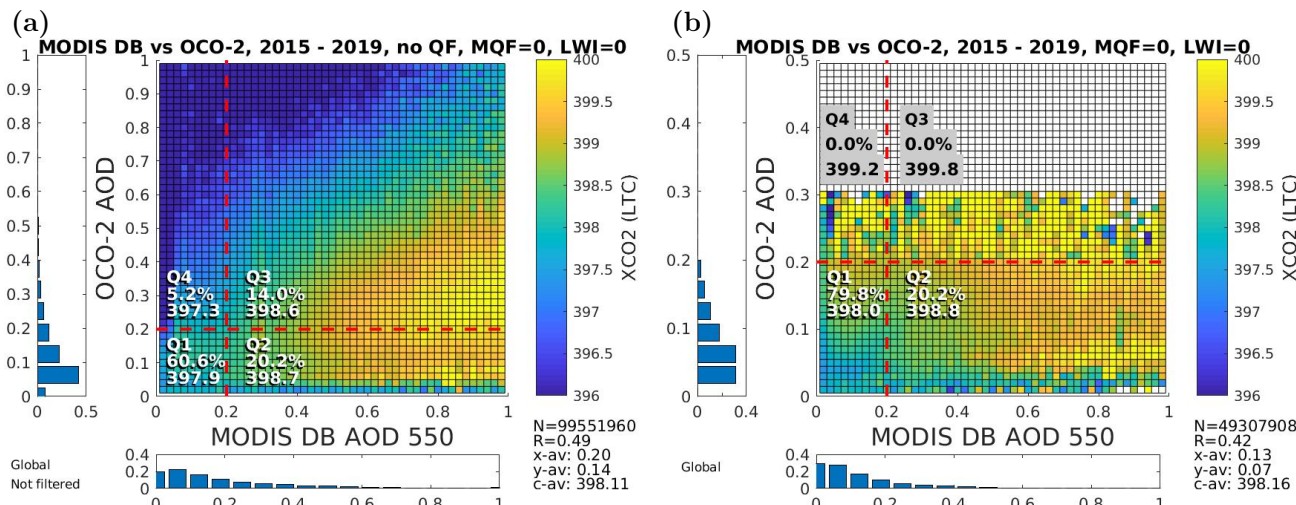

**Figure A6.** Same as Fig. 5 in the paper, but for MODIS DB.

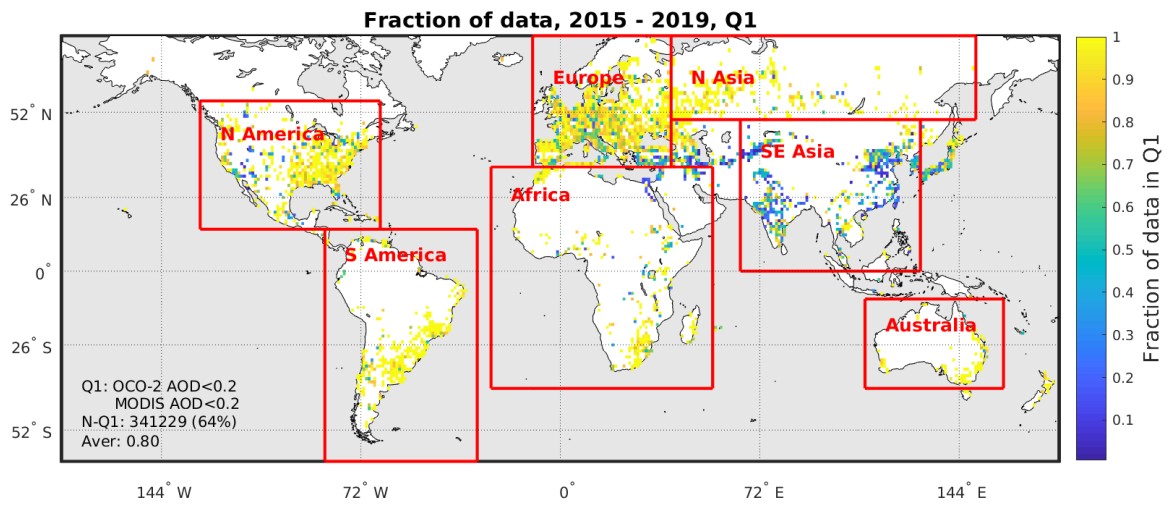

**Figure A7.** Fraction of data in quarter Q1 for urban pixels (areas of dense human habitation using the urban area mask from naturalearth-data.com (NaturalEarth, last access: 22 April 2024). Also shown are the seven geographic areas use in this study.





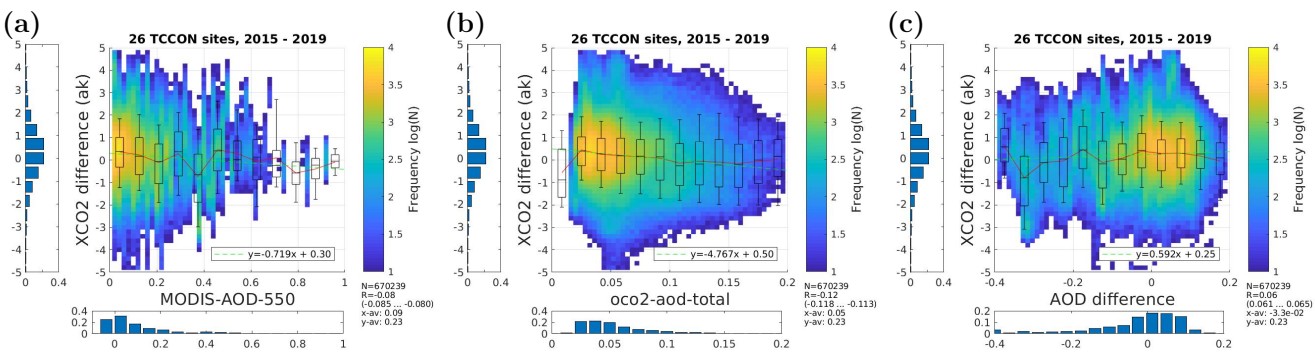

**Figure A8.** XCO$_2$ vs AOD for collocated quality filtered TCCON/OCO-2/MODIS dataset (2015-2019). XCO$_2$ value from TCCON (ttccon-match-60min) is aggregated for one hour time window centered at the OCO-2 overpass time. XCO$_2$ difference is OCO-2 minus TCCON, AOD difference is OCO-2 minus MODIS.

**Figure A9.** Combined bin-averaged plots and linear regression fits for MODIS AOD vs OCO-2 AOD for different subsets of the quality filtered five year collocated dataset. Solid lines show bin averaged OCO-2 AOD values for MODIS AOD bins, dotted lines show the part for bins with very few data (less than 1% of each subset). Dashed lines show the bivariate linear fits (with corresponding colors). (Plots for the two TCCON sets overlap, AOD data is the same.)





**Figure A10.** Same as above, but without OCO-2 quality filtering.