# Peer review of "A global perspective on CO2 satellite observations in high AOD conditions"

_Atmospheric Measurement Techniques, 2024_

## Author Response (AR1)

In this response file we have collected the point-by-point responses to the comments of all three referees, and describe the changes made to the text due to each point in detail. In addition to the changes due to responses, minor edits (language, typos) have been made to the manuscript.

Referee #1

*The Virtanen et al manuscript entitled "A global perspective on $CO_2$ satellite observations in high AOD conditions" tries to evaluate the OCO-2 product from the point of view of aerosols by the combined analysis of the OCO-2 retrieved $XCO_2$ and AOD, collocated MODIS AOD, and TCCON retrieved $XCO_2$. This manuscript found a systematic difference between AOD retrieved by OCO-2 and MODIS, which can impact the retrieval of $XCO_2$. The dependence of the OCO-2 retrieved XCO2 on the AOD difference was also found. But toward the future CO2M mission, in my opinion, besides simple data coverage variation with different AOD threshold, you should also discuss the quality control with the AOD threshold of 0.5.*

**Reply:** We thank the reviewer for the insightful comments which helped us to improve the manuscript. We have revised the manuscript considering the comments and the questions presented.

*Major comments:*

*You only analyzed the data coverage variation after adjusting the AOD threshold from 0.2 to 0.5. According to your analysis of the relationship between OCO-2 retrieved $XCO_2$ and AOD, the accuracy of the former significantly depend on the uncertainty of the later. You should also discuss how to conduct quality filter, or what the data quality will be for CO2M, which will use 0.5 as AOD threshold.*

**Reply:** Clearly the discussion in section 4.5 was not sufficiently clear, as pointed out by all Referees. We have revised this section to better describe the objective of this part of the work and to clarify the issues raised. Please find a more detailed response in the specific comments below.

*The specific comments are listed below:*

*P2L24: Change the order of the sentences "An essential monitoring component will be the Copernicus Anthropogenic $CO_2$ Monitoring Mission (Meijer et al., 2023)." And "While ground-based greenhouse gas measurements are mainly 25 available in developed countries – with limited coverage and representativeness – satellite-based $XCO_2$ information will be irreplaceable in areas where ground-based measurements are not made."*

**Reply:** We agree that this makes text more fluent and have changed the order.

(The line numbers in our replies refer to the latex difference pdf file.)

L24: We have changed

"An essential monitoring component will be the Copernicus Anthropogenic CO 2 Monitoring Mission (Meijer et al., 2023). While ground-based greenhouse gas measurements are mainly available in developed countries – with limited coverage and representativeness – satellite-based XCO 2 information will be irreplaceable in areas where ground-based measurements are not made."

to

"While ground-based greenhouse gas measurements are mainly available in developed countries – with limited coverage and representativeness – satellite-based XCO 2 information will be irreplaceable in areas where ground-based measurements are not made. An essential monitoring component will be the Copernicus Anthropogenic CO2 Monitoring Mission (CO2M; Meijer et al. (2023))."

*P2L33: Check the format of the citations "(e.g., (Houweling et al., 2015; Crowell et al., 2019))", should they be "first name et al. (year)"?*

**Reply:** We agree.

L34: We have changed "(e.g. (Houweling et al., 2015; Crowell et al., 2019))" to "(e.g., Houweling et al. (2015); Crowell et al. (2019))"

*P3L68: "In the collocated database we include only a limited selection of OCO-2 data fields, but the sounding ID in the combined daily files is equivalent with the original OCO-2 lite files, allowing addition of more data fields in an effective manner." It makes me confused. Please further explain it. What is "sounding ID"? Why can you get "more data fields"?*

**Reply:** Initially, only a few selected variables from the OCO-2 data were included in the collocated database (e.g., XCO2, XCO2_quality_flag), to save space. Later we wanted to test possible effects of additional variables, so the database was constructed in such a way that this could be effectively done. Sounding ID is a unique identifier for each observation.

We now realize that this is a technical detail on a level that is probably not relevant for readers and might just cause confusion. We have removed this part from the text.

L69: We have removed the sentence "In the collocated database we include only a limited selection of OCO-2 data fields, but the sounding ID in the combined daily files is equivalent with the original OCO-2 lite files, allowing addition of more data fields in an effective manner."

*P3L70: "The aerosol parameters of the ACOS algorithm include five scatterers, two cloud types (water and ice), two tropospheric aerosol types and a stratospheric aerosol type (sulfate)." This may should be revised as "The aerosol products of the ACOS algorithm include parameters from five scatterers". Is "five scatterers" refer to "two cloud types (water and ice), two tropospheric aerosol types and a stratospheric aerosol type (sulfate)" If so, this may should be revised as "five scatterers, which are two cloud types (water and ice), two tropospheric aerosol types and a stratospheric aerosol type (sulfate)."*

**Reply: This** sentence was poorly formulated, as noted also by Referee #2. We have clarified the sentence, and it now reads:

"The aerosol parameters of the ACOS algorithm include five scatterers, which are two cloud types (water and ice), two tropospheric aerosol types and a stratospheric aerosol type (sulfate)."

*P3L72: There are five aerosol types mentioned in the brackets. Why did you write "Two representative types of tropospheric aerosols"?*

**Reply:** We have reformulated this.

L73: We have changed

"Two representative types of tropospheric aerosols (dust, smoke, sulfate aerosol, organic carbon, and black carbon) are drawn from a climatology based on location and time (Crisp et al., 2021)."

to

"The two most representative types of tropospheric aerosols out of five possible types (dust, sea salt, sulfate aerosol, organic carbon, and black carbon) are drawn from collocated 3-hourly aerosol fields from Goddard Earth Observing System Model, Version 5, Forward Processing for Instrument Teams (GEOS-5 FP-IT; see Crisp et al. (2021))."

*P3L74: "Atmospheric Carbon Observations from Space (ACOS)". Only use abbreviation is ok.*

**Reply:** We now see that ACOS was defined already on line 43 and have removed the second explanation of the abbreviation.

*P3L73: "From the large number of data products…".*

**Reply:** We have replaced "quantities" with "data products".

*P4L86: Add last accessed date after the website.*

**Reply:** Done.

*P4L106: What are "data fields which are not relevant for this study"?*

**Reply:** This is also a technical detail related to the reduction of the size of the collocated database. As this was explained already on line 88 in section 2.2, we have now removed this part from line 106. The removed MODIS data fields include e.g. viewing geometry and algorithm performance parameters, which were not considered in this study.

L112: We have removed the sentence "The reduced daily MODIS files are first created starting from the L2 MODIS Aqua AOD files (MYD04) over land areas by removing pixels that do not have valid aerosol retrieval results and by dropping some data fields which are not relevant for this study."

*P4L120: Why did you use latitude and longitude as the collocation criteria? Using this criterion, the distance between OCO-2 and TCCON is different at different latitudes.*

**Reply:** It is true that latitude affects the sampling in this method. Initially, this was a crude sampling done as a first step when constructing a database, and the adapted approach is similar to earlier studies focusing on the validation of satellite data (e.g. Wunch et al., 2017). The original geolocation information was included in the database. The idea was that different refined sampling criteria (within the larger crudely sampled area) could be tested easily later. When testing smaller sampling areas around TCCON sites, it turned out that the finer sampling had little effect when considering the full global dataset including all 26 TCCON sites. Hence, we kept the original crude collocation criteria. For more detailed studies the sampling should be revisited.

*P4L128: 0.1◦ is also the latitude and longitude threshold? Please describe it. Why did you not use the same spatial and temporal criteria with TCCON?*

**Reply:** In the case of AERONET, the sampling criterion was a circle of 0.1 deg around the AERONET site, assuming pseudo-cartesian lat/lon coordinates. We have now clarified this in the text.

The practical reason for different sampling for AERONET and TCCON is that the data were taken from existing collocation databases originally made for other purposes. For AERONET sampling we can afford a smaller sampling area due to the abundance of the data, while for TCCON we wanted to maximize the number of matches. In addition, the atmospheric lifetime of the species affects the choice of the co-location method. Different sampling criteria were tested also for AERONET, and we found that the results were not too sensitive to the sampling distance. Please see also the reply to comments for Referee #2 for more details.

L142: We have added the sentence "With the abundance of AERONET sites, we could afford a smaller sampling area than that used for TCCON data."

*P4L129: "The OCO-2 observations are not spatially averaged." Not necessary for AERONET part.*

**Reply:** This sentence refers to the comparison of OCO-2 total AOD component to AERONET AOD. Typically, in satellite vs AERONET comparison the satellite data around the AERONET site is spatially averaged and the AERONET data is temporally averaged around the satellite overpass time, to reduce the random noise (see e.g. Virtanen et al. 2018).

Reference: Virtanen, T. H., Kolmonen, P., Sogacheva, L., Rodríguez, E., Saponaro, G., and de Leeuw, G.: Collocation mismatch uncertainties in satellite aerosol retrieval validation, Atmos. Meas. Tech., 11, 925–938, https://doi.org/10.5194/amt-11-925-2018, 2018.

L139: We have changed

"The OCO-2 observations are not spatially averaged."

to

"The OCO-2 observations within the 0.1◦ radius are included in the comparison individually (no spatial averaging). We note that the comparison statistics are typically affected by the spatial and temporal collocation parameters (see e.g. Virtanen et al. (2018)). Different sampling radii and time windows were tested with a subset of data, with minor effects on the results. With the abundance of AERONET sites, we could afford a smaller sampling area than that used for TCCON data."

*P6L148: What is the specific definition of $XCO_2$ anomaly? How to calculate $XCO_2$ anomaly from the median XCO2? The anomaly was calculated from all data, or for every year? After the calculation, have you removed the anomaly or how to use the anomaly? And you said "This is an alternative way to de-trend the data", when did you use the LTC method and when did you use the $XCO_2$ anomaly? Please supplement this section with more information.*

**Reply:** The first sentence in section 3.6, "The OCO-2 XCO 2 anomaly is calculated for each good quality OCO-2 pixel in the collocated dataset as the difference from the median XCO2 value calculated within 500 km for the corresponding OCO-2 orbit." defines the XCO2 anomaly. It is calculated for each good quality pixel, as the difference of XCO2 value from the spatially and temporally varying median. This reference median for each pixel is calculated from the good quality pixels on the same OCO-2 orbit, within 500 km from the pixel considered.

Since the values used to calculate the median are from the same orbit (within few minutes), using this anomaly is expected to remove both the trend and seasonal effects. The idea is that the yearly increase in CO2 and the seasonal variation are large spatial scale effects which are captured by the 500 km portion of an orbit. When the median value is subtracted, the remaining 'anomaly' part is assumed to contain information on local sources and sinks, while the trend and seasonal effects are (presumably) removed. The XCO2 anomaly is used as such (i.e. we do not attempt to 'remove' the anomaly from XCO2 data).

The XCO2 anomaly is used as a proxy of local CO2 emission and compared with AOD data, to study the covariance of aerosols and CO2 emissions. Since the anomaly data looks rather noisy when aggregated to global maps for five years, we do not show it as a map on the manuscript (see Fig. 1 a) below). Instead, XCO2 anomaly statistics are shown in various tables in the manuscript (e.g. Table 2, Table A2, Table A3). In Table 2, the XCO2 anomaly is larger in AOD quarter Q2 than in Q1, indicating a correlation between local XCO2 enhancement (emissions) and MODIS AOD. Figure 1 b) below illustrates this. The XCO2 anomaly analysis supports our findings in the manuscript and shows essentially the same results as obtained with 'usual' XCO2 data, so they are not shown in more detail in the manuscript.

We have slightly extended the description of XCO2 anomaly in section 3.6 to clarify these points.

[Figure]

Figure 1. a) Local XCO2 anomaly data [ppm] for five years aggregated to 0.5 deg grid cells. b) Correlation observed between XCO2 anomaly and AOD.

L165: We have changed

"The OCO-2 XCO2 anomaly is calculated for each good quality OCO-2 pixel in the collocated dataset as the difference from the median XCO2 value calculated within 500 km for the

corresponding OCO-2 orbit. This is an alternative way to de-trend the data, instead of applying the simple LTC. Unlike LTC, the anomaly method also effectively de-seasonalises the data. It also allows to study the covariance of AOD values and local XCO2 anomalies caused by possible CO2 sources and sinks."

to

"The OCO-2 XCO2 anomaly is calculated for each good quality OCO-2 pixel in the collocated dataset as the difference from a local, temporally varying median value. This median is calculated from the good quality pixels in the same OCO-2 orbit, within 500 km from the pixel considered. The idea is that the yearly increase in CO2 and the seasonal variation are large spatial scale effects which are captured by the 500 km portion of an orbit. When the median value is subtracted, the remaining 'anomaly' part is assumed to contain information on local sources and sinks, while the trend and seasonal effects are removed. This is an alternative way to de-trend the data, instead of applying the simple LTC. Unlike LTC, the anomaly method also effectively de-seasonalises the data. It also allows to study the covariance of AOD values and local XCO 2 anomalies caused by possible CO2 sources and sinks. While most of the results shown in this work have been processed with the linear trend correction, the corresponding XCO2 anomaly results are also shown where appropriate to support the analysis."

*P6L167: The dust loads may be identified by the areas, but "biomass burning aerosols increase AOD in central Africa and South-East Asia" lack of evidence.*

**Reply:** We think that the high AOD values in these areas are often contributed to biomass burning, see e.g. van der Werf et al. (2010), Myhre et al. (2009), or Kinne (2019). As this is not in the focus of the manuscript, we have removed the reference to biomass burning aerosols.

References:

Van der Werf et al., 2010, www.atmos-chem-phys.net/10/11707/2010/

Myhre et al., 2009, www.atmos-chem-phys.net/9/1365/2009/

Kinne, 2019, https://doi.org/10.1080/16000889.2019.1623639

L190: We have removed the text ", biomass burning aerosols increase AOD in central Africa and South-East Asia,".

*P6L172: I think the largest differences appear in Central Asia and South Asia.*

**Reply:** Our definition of 'South-East Asia', as shown in Fig. A7, is rather loosely defined. It covers areas usually understood as South Asia, East Asia and parts of South-East Asia.

L195: We have changed

"The largest differences in AOD appear to be concentrated largely in the high AOD areas in South-East Asia, where OCO-2 AOD is lower than MODIS AOD."

to

"The largest differences in AOD appear to be concentrated largely in the high AOD areas in parts of Asia, where OCO-2 AOD is lower than MODIS AOD."

*P6L173: "These positive difference values are related to the MODIS DT algorithm permitting small negative AOD values." Any references?*

**Reply:** The negative MODIS AOD values are discussed in Sayer et al. (2014). This reference was given after a couple of sentences on line 176. We have moved the citation to line 173.

*P7 Fig.1: The colorbar of panel (b) should be adjusted because 0 and invalid data are both white.*

**Reply:** We have duly adjusted the color for missing data in Fig. 1 b).

*P8L187: You should point out earlier.*

**Reply:** The discussion on the wavelength issue is lengthy, which is why it was delayed until this point. We added a note on it at the beginning of section 4.1, with reference to later discussion.

L193: We have added the sentence "Note that the OCO-2 AOD is retrieved at 755 nm, while the MODIS AOD is obtained at 550nm; the effect of the wavelength difference will be discussed below."

*P8L200: You acquired 770 nm AOD by averaging 675 nm AOD and 870 nm AOD. Can you really achieve that by the simple average? Please give more evidence or reference.*

**Reply:** A simple average is not the optimal way to scale the wavelengths but should be acceptable in this case. Using scaling with Ångström exponent with a smaller subset of data did not change the results dramatically. Please see comments to Referee #2 for more detailed discussion.

We have added a note on this to section 3.3.

L144: We have added the sentence "While this simple approach may not be the most accurate, it is sufficiently accurate for our purposes. A more accurate method for the wavelength scaling using the Ångström exponent from AERONET was tested for a subset of data, and we did not find significant differences in the results."

*P8L210: How did you define urban areas?*

**Reply:** The urban areas are discussed from line 329 on, where references are given. We have added a note on this earlier to L210.

L244: We have added the note "(see Fig. A7 for definition of urban areas)".

*P9L217: Please point out the spatial resolution of OCO-2 again.*

**Reply:** We have added the OCO-2 resolution (approx 1x2km²) to the text here.

L254: We have added the text "(approximately 1×2 km²)".

*P10L250: How did you handle with the AOD exactly equal to 0.2?*

**Reply:** There were practically no data points with AOD exactly at 0.2. They are included in the larger AOD bin. We have changed the text to replace "AOD<0.2" by "AOD<=0.2".

*P13Figure 4 (b): You should distinguish 0 values and invalid values. Both of them are white.*

**Reply:** We have duly changed the color for missing data.

P13L299: Table A3 or Figure A3?

**Reply:** The reference was incomplete. We have changed this to "(see Table A3)".

*P14L316: What does "the measured $CO_2$ absorption is divided into too short distance" mean?*

**Reply:** The measured radiance at CO2 absorption channels carries information of the total column load of CO2 in the observed light path. The inversion of measured radiance signal to CO2 column load (XCO2) requires information on the light path length. The light path length is affected by aerosols: more aerosol means effectively more scattering, and a longer light path. If the retrieved AOD is incorrect, the light path length is also incorrect, and this directly affects the retrieved XCO2 value. We have revised the text to state this more explicitly.

L375: We have replaced

"If the aerosol load is underestimated in the retrieval (Q2), the light path is also underestimated, and the measured CO2 absorption is divided into too short distance, leading to overestimation of XCO2. Similarly, if AOD is overestimated, the light path is also overestimated, causing underestimation of XCO2."

by

"The top of atmosphere radiance measured by OCO-2 contains information on the total amount of CO2 along the light path, and inversion of this information to XCO2 values requires knowledge of the light path length, which is affected by aerosols. If the aerosol load is overestimated in the retrieval (Q4), the light path is also overestimated, and the measured CO2 absorption is divided into too long distance, leading to underestimation of XCO2. Similarly, if AOD is underestimated (Q2), the light path is also underestimated, causing overestimation of XCO2."

*P15L340: Again, if you aquire OCO-2 $XCO_2$ anomaly by calculating the difference between the OCO-2 $XCO_2$ data and the median value along 500 km orbit, how can the anomaly be used to de-seasonalize and de-trend the data? Please describe more details about this.*

**Reply:** The XCO2 anomaly values are used as such, instead of using them to 'correct' the XCO2 values. It is assumed that the large scale seasonal and trend effects are removed when subtracting the 500 km median, and that the anomaly contains information on the XCO2 variability in the local spatial scale, including local sources (and sinks). Please see the extensive reply to the previous similar comment above (reply to comment P6L148).

*P15Table2: "TCCON(1)" should be revised as "OCO-2(1)"?*

**Reply:** The naming refers to different subsets of OCO-2 data, hence 'global', 'urban' and 'TCCON'. To distinguish between the two data sources (OCO-2 and TCCON) for the collocated OCO-2 & TCCON dataset, we have used the numbers in parenthesis, (1) and (2). We have revised the Table caption to clarify this.

Table 2 caption: We have replaced "For collocated TCCON data two XCO2 values are given, from OCO-2 (1) and from TCCON (2), respectively." with "Three datesets are used: global, urban and one collocated with TCCON. For the collocated TCCON data two XCO2 values are given, from OCO-2 (labeled TCCON(1)) and from TCCON (labeled TCCON(2)), respectively."

*P16L365: "OCO-2 seems to slightly overestimate XCO2 for low AOD values, and underestimate at high AOD values."*

*-For panel (a) of Figure A8, you used MODIS AOD and XCO2 difference (deviation of OCO-2 from TCCON). MODIS AOD and TCCON XCO2 can be considered as references, so you can definitely get the statement about overestimate or underestimate of OCO2 XCO2 at different AOD.*

**Reply:** We have revised the text.

L429: We have replaced "OCO-2 seems to slightly overestimate XCO2 for low AOD values, and underestimate at high AOD values" with "OCO-2 slightly overestimates XCO2 for low AOD values, and underestimates at high AOD values".

*-For panel (c), it seems that the XCO2 difference is close to zero when the AOD difference is in the range of 0 to 0.1, and when the AOD difference becomes higher or lower, the XCO2 difference will be minus, which means the underestimate of OCO2 XCO2. It seems to be not totally consistent with the statement of Figure 5 "If the aerosol load is underestimated in the retrieval (Q2), the light path is also underestimated, and the measured CO2 absorption is divided into too short distance, leading to overestimation of XCO2. Similarly, if AOD is overestimated, the light path is also overestimated, causing underestimation of XCO2." Do you have any comments on it?*

**Reply:** The discussion at Figure 5 was meant as the first impression and possible explanation of the results seen in Fig. 5 a), accompanied by the disclaimer that it is not easy to distinguish the two effects of co-emission of aerosols and CO2 and the aerosol bias on XCO2 retrieval. The discussion of the light path is a possible explanation, which turned out not to be consistent with the results obtained with collocated TCCON data, as correctly pointed out by the reviewer.

We find that it is difficult to draw detailed conclusions from Fig. A8 c), because there are two dependencies mixed in the plot. First, the AOD difference between OCO-2 and MODIS depends on the MODIS AOD in a nontrivial way as shown in Fig. 2, with OCO-2 low bias at one end and high bias at the other. Second, the XCO2 bias depends also on MODIS AOD. We find that Fig. A8 a) best describes what we can say about the XCO2 bias, but nevertheless we wanted to show Fig. A8 c) for completeness.

It is certainly important to state these points explicitly in the manuscript. We have revised the discussion to section 4.3 to achieve this.

L374: We have replaced

"The striking connection between XCO2 and the relative AOD values between the two instruments in Fig. 5 a) can potentially be explained by the light path length used in the ACOS full physics retrieval."

by

"As a first guess, the striking connection between XCO2 and the relative AOD values between the two instruments in Fig. 5 a) could potentially be explained by the light path length used in the ACOS full physics retrieval."

L382: We have added the sentence: "However, for Q2 the interpretation turns out to be more complicated, when the reference XCO 2 data from TCCON is considered, as discussed below."

L430: We have added the text: "As with Fig. 7 c), the interpolation of Fig. A8 c) is complicated, since there are two aerosol related dependencies affecting the data. First, the AOD difference between OCO-2 and MODIS depends on the MODIS AOD in a nontrivial way as shown in Fig. 2, with OCO-2 low bias at one end and high bias at the other. Second, the XCO 2 bias depends also on MODIS AOD."

*P18L388: The XCO2 and AOD also show negative correlation in North America and Europe. Do you have any comments on this?*

**Reply:** Here we wanted to highlight only the most obvious differences between the areas. In Fig. 8 c) the negative slope of XCO2 as function of AOD is most obvious for North Asia, but the linear fit slopes are negative also for Europe and South America (for North America the slope is slightly positive). Looking at Fig. 4 b), we see that in North America there are areas of negative and positive correlation, but the data is noisy. For South America the correlation is weaker, and close to zero in large areas.

Actually, most of the areas have negative correlation (when looking at the matrix plots, not shown in the manuscript), with the exception of Africa and 'South-East' Asia.

We have reformulated the text.

L465: We have replaced

"Most of the areas show a positive correlation between XCO2 and AOD, with the clear exception of Northern Asia, which is dominated by strong seasonal cycle of XCO2 . We also note that the northern areas are strongly undersampled in winter months due to snow cover and high SZA, which prevent MODIS aerosol retrievals."

by

"SE Asia and Africa show a positive correlation between XCO2 and AOD, N America, S America, and Australia have slopes closer to zero, while Europe and in particular Northern Asia have negative slopes. The northern areas are dominated by strong seasonal cycle of XCO2 , and are strongly undersampled in winter months due to snow cover and high SZA, which prevent MODIS aerosol retrievals."

*P19L395: "Satellite XCO$_2$ retrievals are known to have higher uncertainty in high aerosol conditions." Where is this statement summarized from? From Figure A8 (a) and (b), it can be seen that the OCO-2 XCO2 also has deviation from TCCON XCO2 at low AOD.*

**Reply:** This sentence refers to earlier research in the field; we have added references to Connor et al. (2016) and O'Dell et al. (2018). Figure A8 a) and b) are plotted using quality filtered OCO-2 data, where high AOD cases (OCO-2 AOD over 0.2) have already been removed. Unfiltered OCO-2 data would show a larger spread (see Fig. 2 below). However, it is true that there is considerable spread in the data also at the low AOD region.

[Figure]

Figure 2. Same as Fig. A8 a) and b) in the manuscript, but without OCO-2 quality filtering.

L473: We have added the references "(Connor et al., 2016; O'Dell et al., 2018)".

*P21L427: For future CO2M, you found the large increase of data coverage after adjusting AOD threshold from 0.2 to 0.5. After adjusting, there will be more underestimated and overestimated data when the AOD is below 0.5, i.e., the data in Q1 area you defined will include data in Q2 and Q4 areas. You should discuss how to conduct quality filter for CO2M, or the data quality with the AOD threshold of 0.5.*

Clearly the discussion in section 4.5 was not sufficiently clear, as pointed out by all referees. The idea here is that the coming CO2M mission will have a dedicated aerosol instrument - Multi-Angle Polarimeter (MAP) – which is expected to allow XCO2 retrieval at heavier aerosol conditions, with AOD threshold of 0.5.

Here we use the collocated, quality filtered OCO-2/MODIS dataset as a proxy for CO2M data. This dataset includes high MODIS AOD pixels, although the OCO-2 quality filter including an AOD threshold of 0.2 has been applied. We assume that the OCO-2 quality filtering assures that the XCO2 data is of good quality even for higher MODIS AOD cases, as CO2M data is expected to be up to AOD of 0.5. We further assume that the MODIS AOD in the collocated dataset is representative of 'true' AOD and can be used to study the AOD threshold, even though the OCO-2 quality filtering has removed a large part of the original pixels.

With this collocated data set, we can test what is the effect of relaxing MODIS AOD threshold from 0.2 to 0.5. Note that this does not mean that we extend the OCO-2 coverage; the MODIS AOD threshold used here is an additional constraint on the quality filtered OCO-2 data.

In the manuscript we did not address the effect of different MODIS AOD thresholds on the XCO2 retrieval quality. Figure 3 below shows such assessment: We find that in this case the MODIS AOD threshold did not have any significant effect on the XCO2 retrieval quality. But as mentioned, in these cases we use the quality filtered OCO-2 data, which should be of sufficiently good quality. Having said that, we need to clarify that the purpose of this study was not to improve the capability of OCO-2 to deal with higher aerosol loads, but to find how increasing the AOD threshold affects the coverage.

[Figure]

Figure 3. Effect of MODIS AOD threshold on XCO2 comparison between OCO-2 and TCCON. The OCO-2 quality flag has been applied.

We do not propose replacing the OCO-2 quality filtering with a single MODIS AOD threshold. The OCO-2 quality filter is based on 32 different tests, of which the total AOD threshold of 0.2 is only one component. Replacing this with a MODIS AOD threshold, even as stringent as 0.2, would result in reduced XCO2 quality, as shown in Figure 4 below.

[Figure]

Figure 4. Effect of MODIS AOD threshold on XCO2 comparison between OCO-2 and TCCON, when the OCO-2 quality flag has not been applied.

In conclusion, here we treat the quality filtered OCO-2 data as a proxy of the coming CO2M data, which can be further filtered by using MODIS AOD thresholds. Our main finding, then, is that if CO2M can handle AODs up to 0.5, this will increase coverage, particularly in urban areas, compared to a case where AOD only up to 0.2 could be allowed.

We have clarified the text in section 4.5 to bring these points explicitly out.

L477: We have added the sentence: "The CO2M mission will include a dedicated aerosol instrument, the Multi-Angle Polarimeter (MAP), and is expected to be better equipped to deal with high aerosol conditions."

L480: We have added the text:

"In this section we estimate how the selected AOD threshold affects the coverage of satellite XCO 2 retrievals, in particular in urban areas with high co-emission of aerosols and CO2.

Here we use the collocated, quality filtered OCO-2/MODIS dataset as a proxy for CO2M data. This dataset includes high MODIS AOD pixels, although the OCO-2 quality filter including an AOD threshold of 0.2 has been applied. We assume that the OCO-2 quality filtering assures that the XCO 2 data is of good quality even for higher MODIS AOD cases, as CO2M data is expected to be up to AOD of 0.5. This assumption is supported by a comparison of quality filtered OCO-2 XCO 2 data against TCCON, where additional collocated MODIS AOD thresholds had minimal effect on the retrieval quality (not shown). We further assume that the MODIS AOD in the collocated dataset is representative of 'true' AOD and can be used to study the AOD threshold, even though the OCO-2 quality filtering has removed a large part of the original pixels. With this collocated data set, we can test what is the effect of relaxing MODIS AOD threshold from 0.2 to 0.5. We emphasise that this does not mean that we extend the OCO-2 coverage (or propose to relax the OCO-2 AOD threshold); the MODIS AOD threshold used here is an additional constraint on the quality filtered OCO-2 data."

L504: We have added the text:

"In conclusion, here we have used the quality filtered OCO-2 data as a proxy of the coming CO2M data, which can be further filtered by using AOD thresholds from collocated MODIS data. We find that if CO2M can handle AODs up to 0.5, this will significantly increase the coverage, in particular in the urban areas, compared to a case where AOD only up to 0.2 could be allowed. We also find that due to the correlation found between AOD and XCO2, including data with higher AOD increases the mean XCO 2 values, especially for the urban pixels."

Referee #2

*I understood that this manuscript studies the possibility of CO2 retrieval in high AOD pixels by the AOD threshold change.*

*Because of the disadvantage of spatial coverage for CO2 satellites, this study will contribute to enhancing the global coverage of observation data for CO2 satellites. Although the purpose of the manuscript is acceptable, the detailed analysis and results are not so clear. For details..*

**Reply:** We thank the referee for the valuable comments. We have revised the manuscript accordingly, to make the analysis and results clearer.

*1) Section 2: The study used the Dark Target (DT) algorithm to concentrate the urban surface. However, the DT algorithm have large uncertainty of AOD over land surface as compared to the ocean surface. For the detailed analysis of retrieval uncertainty related to the AOD, retrieval results over land surface are carefully handled. Do you have the same results when AOD from MODIS uses the Deep-blue algorithm?*

**Reply:** As discussed in the manuscript, we have repeated the analysis using MODIS Deep Blue (DB) algorithm results, and the results are shown in the Appendix. While the global patterns of AOD (difference) are somewhat different than with MODIS Dark Target (DT), we find that the global statistics and conclusion regarding the connection to XCO2 retrievals drawn from the data are largely the same. We have added a note on this in Section 2.2. MODIS DB results are shown in Fig. A2 and Fig. A6 in the Appendix.

L86: We have added the sentence: "While the global AOD patterns are somewhat different between DT and DB, we find that the global statistics and conclusion regarding the connection to XCO2 retrievals are largely the same."

*2) L72-L73: For the OCO-2 AOD retrieval, two representative types selection is confused. Does this sentence mean that spatio-temporal variated climatological types are selected for the AOD retrieval? In addition, Is the AOD from OCO-2 hard to consider the 'case dependent' aerosol types?*

**Reply:** This sentence was poorly formulated, as noted also by Referee #1. We have clarified the sentence.

L71: We have replaced

"The aerosol parameters of the ACOS algorithm include five scatterers, two cloud types (water and ice), two tropospheric aerosol types and a stratospheric aerosol type (sulfate). Two representative types of tropospheric aerosols (dust, smoke, sulfate aerosol, organic carbon, and black carbon) are drawn from a climatology based on location and time (Crisp et al., 2021)."

by

"The aerosol parameters of the ACOS algorithm include five scatterers, which are two cloud types (water and ice), two tropospheric aerosol types and a stratospheric aerosol type (sulfate). The two most representative types of tropospheric aerosols out of five possible types (dust, sea salt, sulfate aerosol, organic carbon, and black carbon) are drawn from collocated 3-hourly aerosol fields from

Goddard Earth Observing System Model, Version 5, Forward Processing for Instrument Teams (GEOS-5 FP-IT; see Crisp et al. (2021))."

*3) L99: For the AERONET AOD reference, Eck et al. (1999) is too old to explain the Version 3. Giles et al. (2019) or Sinyuk et al. (2020) are more suitable.*

*Giles, David M., et al. "Advancements in the Aerosol Robotic Network (AERONET) Version 3 database–automated near-real-time quality control algorithm with improved cloud screening for Sun photometer aerosol optical depth (AOD) measurements." Atmospheric Measurement Techniques 12.1 (2019): 169-209.*

*Sinyuk, Alexander, et al. "The AERONET Version 3 aerosol retrieval algorithm, associated uncertainties and comparisons to Version 2." Atmospheric Measurement Techniques 13.6 (2020): 3375-3411.*

**Reply:** We thank the referee for pointing out this omission. We have checked for more recent references and added them to the manuscript.

L107: We have added the reference "Sinyuk et al. (2020)".

L108: We have added the reference "(Giles et al., 2019)".

*4) Section 3.1: For the collocation, I can't find the cloud screening method before the grinding. The AOD retrieval products are very important to the cloud screening before the analysis, although respective AOD retrieval algorithm have there own cloud masking method.*

**Reply:** Both OCO-2 and MODIS data are cloud screened, using their own respective methods, before the collocation. The MODIS Collection 6 cloud mask used in aerosol retrieval is described in Levy et al. (2013) and references therein and has been continuously developed for aerosol retrieval purposes. The MODIS quality flag is further used to reduce possible cloud contamination e.g. by thin cirrus. OCO-2 data are cloud-screened before the Level-2 retrieval. The cloud screening is described and validation shown in Taylor et al. (2016).

We hope that the combined use of cloud masks from both instruments helps mitigate cloud contamination issues. However, we are aware that cloud screening is a trade-off between coverage and possible residual clouds contaminating the retrieval results. High AOD outliers seen in the data may be caused by insufficient cloud screening.

The effect of cloud screening on the collocated dataset is indicated in the manuscript in section 3.1 (and Table A1 and Fig. A1), but not explicitly discussed. We have added a brief note on the effect of cloud screening (and possible residual clouds) in section 3.1.

References:

Levy, R. C., Mattoo, S., Munchak, L. A., Remer, L. A., Sayer, A. M., Patadia, F., and Hsu, N. C.: The Collection 6 MODIS aerosol products over land and ocean, Atmospheric Measurement Techniques, 6, 2989–3034, https://doi.org/10.5194/amt-6-2989-2013, 2013.

Taylor, T. E., O'Dell, C. W., Frankenberg, C., Partain, P. T., Cronk, H. Q., Savtchenko, A., Nelson, R. R., Rosenthal, E. J., Chang, A. Y., Fisher, B., Osterman, G. B., Pollock, R. H., Crisp, D., Eldering, A., and Gunson, M. R.: Orbiting Carbon Observatory-2 (OCO-2) cloud screening

algorithms: validation against collocated MODIS and CALIOP data, Atmos. Meas. Tech., 9, 973–989, https://doi.org/10.5194/amt-9-973-2016, 2016.

L124: We have added the sentence "We note that although both data products are cloud screened, possible mutual cloud contaminated pixels can cause erroneous high AOD values, which may affect the obtained correlation coefficients."

*5) Section 3.3 and more: For the collocation, the author has to check the consistency of data variability within the collocation range (spatially and temporally). Could you provide the reference for spatial and temporal collocation ranges?*

**Reply:** We are aware of the collocation mismatch uncertainty related to comparing point-like AERONET observations to snapshot type satellite aerosol data over larger areas. The sampling distance used for satellite data and temporal averaging window size for AERONET data affect the comparison metrics (see e.g. Virtanen et al. 2018).

In this work, we content ourselves with a limited comparison with AERONET, using a simple sampling strategy as described in Section 3.3. This is mainly done to confirm the results we see in the comparison with MODIS: that the OCO-2 AOD has a slope around 0.3 with respect to MODIS AOD, and a slope around 0.4 when accounting for the wavelength difference. Based on Virtanen et al., 2018 (e.g. Figure S8 in the Supplement), a sampling distance of 0.1 degree and temporal window of one hour seem reasonable. We have also tested AERONET comparison with a subset of data using different collocation parameters. We note that the MODIS AOD product, used as the reference for OCO-2 here, has been extensively validated elsewhere (e.g. Levy et al. (2013)), and we do not intend to repeat the effort here with the limited collocated dataset.

[Figure]

Figure 1. Effect of different collocation criteria on the AERONET comparison. The results differ only slightly when using a) distance of 0.1 degree for averaging OCO-2 data around the AERONET site and a temporal averaging window of ±30 min centered at the OCO-2 overpass time for AERONET data; or b) distance of 0.5 degree and a temporal averaging window of ±60 min.

We have added discussion on the spatial and temporal collocation ranges to Section 3.3.

Reference:

Virtanen, T. H., Kolmonen, P., Sogacheva, L., Rodríguez, E., Saponaro, G., and de Leeuw, G.: Collocation mismatch uncertainties in satellite aerosol retrieval validation, Atmos. Meas. Tech., 11, 925–938, https://doi.org/10.5194/amt-11-925-2018, 2018.

L140: We have added the text "We note that the comparison statistics are typically affected by the spatial and temporal collocation parameters (see e.g. Virtanen et al. (2018). Different sampling radii and time windows were tested with a subset of data, with minor effects on the results."

*6) Section 4: Did only use the Quality flag value from OCO-2? Why don't you use the quality flag for another satellite platform (MODIS AOD)?*

**Reply:** We did use the quality flag from MODIS as well. We systematically removed the lowest quality MODIS pixels (MODIS quality flag 0), as discussed in Section 2.2. The OCO-2 quality flag is discussed more, since the focus is more on the AOD threshold used in that flag. We have added clarification of the quality flags used in the text in Section 2.2 and Section 4.

L94: We have added the text "Note that the MODIS quality flag is systematically applied throughout the results in this paper, while the use of OCO-2 quality flag varies. In the rest of the paper, when the use of quality flag or quality filtering is discussed, this refers to the OCO-2 quality flag."

L189: We have replaced "(using the OCO-2 quality flag)" by "(the MODIS quality flag is always applied; here we use also the OCO-2 quality flag)".

*In most of the results' subsections, the paper did not have sub summary or sub-conclusion. For this reason, it may be difficult to a connection among the results.*

**Reply:** We agree that the purpose of different subsections and the conclusions drawn from each section were not described clearly enough. We have added more discussion to the text, including conclusions to section 4.1, 4.2, and 4.3, and introductory parts to sections 4.2 and 4.4.

L266: We have added the text: "To conclude, in this section we considered the differences between collocated the MODIS DT AOD product and the OCO-2 total AOD component. We find that the AOD difference depends on region. OCO-2 tends to overestimate the aerosol load in regions with low MODIS AOD. More important for the XCO2 retrievals, OCO-2 tends to severely underestimate AOD in the high MODIS AOD regions (including areas with high anthropogenic emissions), which may have an effect on the XCO2 retrievals in these regions."

L272: We have added the text: "In this section we will compare the OCO-2 total AOD component to MODIS AOD statistically for the full collocated dataset using e.g. density scatter plots. Specifically, we address the question of how well the OCO-2 quality filtering works from the point of view of aerosols. The OCO-2 quality filter uses an AOD threshold of 0.2, among several other tests, to remove heavy aerosol conditions. We use collocated MODIS AOD data to assess the performance of the OCO-2 AOD filter."

L341: We have added the text: "To conclude this section, we have found that the quality filtered OCO-2 data contains a large fraction of data with high MODIS AOD, potentially affecting the XCO2 retrieval quality. These data are more frequent in densely populated areas with high aerosol and CO2 emissions. Hence, for monitoring anthropogenic CO2 emissions with satellites, it is crucial that the high AOD cases are carefully detected and treated in the satellite retrievals."

L446: We have added the text: "To conclude this section, we find that there is a linear relation between OCO-2 XCO 2 and MODIS AOD (Fig. 6 a). We also find a linear relation between the OCO-2 XCO 2 bias and MODIS AOD (Fig. A8 a). We also find a relation between the AOD difference between OCO-2 and MODIS and the OCO-2 XCO2 values, as shown in Fig. 8 a). Aerosols are related to OCO-2 XCO 2 retrievals in two ways: there is a real correlation between XCO2 and AOD, due to co-emission of aerosols and CO2. There is also an aerosol related bias in the OCO-2 retrievals, which acts in opposite direction than the co-emission but with smaller magnitude, thus partly masking the co-emission effect. However, we are unable to directly relate the AOD difference observed between OCO-2 and MODIS to the XCO2 difference observed between OCO-2 and TCCON for the quality filtered data. This is due to the non-trivial AOD difference observed between OCO-2 and MODIS, further complicating the entanglement caused by the two competing aerosol effects."

*7) L184-L185: How to eliminate the cloud contamination? Cloud contamination affects the high AOD, and it affects the high correlation between OCO-2 and MODIS, when both algorithms have cloud contamination.*

**Reply:** As discussed above at point 4), we make an effort to screen for clouds. Cloud contamination is one of the likely reasons for high AOD outliers for each instrument. It is possible that simultaneous cloud contamination of both instruments affects the correlation data. However, the vast majority of data is at the low AOD range, as seen in Fig. 2. Studying the effect of residual clouds in more detail would require some reference data of the actual cloud cover and is beyond the scope of this work. We have added a note of the effect of possible simultaneous cloud contamination to the observed correlations to Section 4.1.

L212: We have added the text "We note that possible mutual cloud contamination of collocated data points could lead to erroneous high AOD values for both instruments, possibly leading to higher correlation values than without cloud contamination. However, data from each satellite is cloud screened with their respective cloud masks, and the vast majority of data is in the low AOD region, reducing the probability of large bias."

*8) L194: I don't agree with the spectral conversion based on the MERRA-2. To use this method, the author has to analyze the intercomparison between MERRA-2 angstrom exponent and AERONET angstrom exponent.*

**Reply:** We realize that the use of MERRA-2 Ångström exponent for the spectral conversion involves high uncertainty. This is why we chose to compare the AOD results at their original wavelengths, 550 nm for MODIS and 755 for OCO-2, for most of the paper, and address the wavelength difference respectively. As we discuss in the manuscript, we expect that part of the difference seen between the instruments is due to the wavelength difference. MERRA-2 data was chosen, because it is readily available for the full global collocated MODIS/OCO-2 dataset.

We use the spectral conversion with MERRA-2 data merely to get a rough estimate on the effect of the wavelength difference on the AOD difference. This is done only in a statistical sense for the global dataset, understanding that the high uncertainties involved with the scaling do not allow for a more detailed comparison. The main conclusion drawn from this is that while the slope of OCO-2 AOD against MODIS AOD is ~0.3 before spectral scaling, it is ~0.5 after the scaling, i.e. the wavelength difference explains part, but not all, of the difference.

We agree that the spectral conversion using AERONET data has much lower uncertainty. To support the MERRA-2 exercise, we compared OCO-2 AOD at 755 nm with the AERONET AOD data at different wavelengths in Fig. A5 in the manuscript. The AERONET results confirm that the OCO-2 AOD (at 755 nm) slope against AERONET AOD is ~0.3 when using 550 nm and ~0.5 when using average of 675 nm and 870 nm for AERONET.

To further support this analysis, we have now repeated the analysis using a subset of collocated MODIS/OCO-2/AERONET data. Here we have used the AERONET Ångström exponent to do the spectral conversions. Here we also use a slightly different approach than in Fig. A5, in that we average the OCO-2 data within 0.125 deg around the AERONET site and average the AERONET data within 1 hour of the overpass time. From Fig. 2 below we see that comparison of OCO-2 AOD to AERONET AOD scaled to 550 nm and to 760 nm, respectively, gives similar results (in terms of the slope) as in Fig. A5 of the manuscript. The small difference in the numbers may be explained by the smaller subset of data used here.

[Figure]

Figure 2. Comparison of OCO-2 AOD at 760 nm against AERONET AOD at 550 nm (left) and against AERONET AOD at 760 nm (right). The AERONET AOD values are scaled to different wavelengths using the AERONET Ångström exponent.

Figure 3 below shows a comparison similar to Fig. A4 in the manuscript, where OCO-2 AOD is compared to MODIS AOD at 550 nm, first using the original data at wavelength of 760 nm and then using OCO-2 AOD scaled to 550 nm, this time using the Ångström exponent from AERONET. Naturally, here this is done only for the subset of data collocated with AERONET. The results agree strikingly well with those shown in Fig. A4 in the manuscript. The slope obtained here is 0.31 and 0.33 in Fig. A4 when using the original wavelengths, and 0.47 in here and in Fig. A4 when using the OCO-2 AOD scaled to 550 nm.

[Figure]

Figure 3. Comparison of OCO-2 AOD to MODIS AOD using the original wavelengths (left), and using AERONET Ångström exponent to scale OCO-2 AOD to 550 nm (right).

Based on this further analysis, we suggest that the use of MERRA-2 Ångström data for the spectral conversion is sufficient for the limited purpose of assessing the effect of wavelength difference to the AOD comparison. We have added a note summarizing this further analysis to Section 4.1.

L228: We have added the text "The use of MERRA-2 data potentially induces high uncertainty to the spectral conversion. We use this method merely to get a rough estimate of the effect of the wavelength difference on the AOD difference. This is done only in a statistical sense for the global dataset, understanding that the high uncertainties involved with the scaling do not allow for a more detailed comparison. The main conclusion drawn from this is that while the slope of OCO-2 AOD against MODIS AOD is 0.3 before spectral scaling, it is 0.5 after the scaling, i.e. the wavelength difference explains part, but not all, of the difference. The spectral conversion was repeated with a smaller subset of data using Ångström exponent from AERONET, and the results largely agreed with the global dataset."

*9) Section 4.2: I clearly don't know the purpose of this section. Only quality checking? or making the threshold of AOD to define the high AOD?*

**Reply:** This section has two purposes: 1) it continues the AOD comparison between OCO-2 and MODIS in a statistical sense (while section 4.1 concentrated on the spatial differences); 2) it assesses the AOD threshold of 0.2 currently used in OCO-2 retrievals to designate the good quality retrievals. While the first point can be considered as 'quality checking', the second point aims at discovering how well the current OCO-2 quality filtering works from the point of view of aerosols (there are other contributing factors, not addressed here). The division to four 'AOD quarters' in the density scatter plots shows the cases where the AOD-based quality filter works well (Q1, where both instruments agree that AOD is low, and Q3, where instruments agree that AOD is high), and where it could be improved (Q2, where high AOD cases may be included to good quality data, and Q4 where low AOD cases may be unnecessarily removed).

We have added a brief introduction to section 4.2 to make the purpose of the section more explicit. (See point 6 above for the description of changes made in the text.)

*In addition, for the AOD quality checking, gridded dataset is not adequate. If you use the gridded AOD data, the author has to make a finer resolution.*

**Reply:** Figure 2 shows clearly the distribution of AOD differences between the two instruments. The main findings here are (1) the distribution of data in the four AOD quarters, and (2) the bias of OCO-2 AOD as function of MODIS AOD (with similar behavior observed in comparison against AERONET in the Appendix). This is shown both as a linear fit, and as binned means with respect to MODIS AOD bins. These results do not change when higher resolution is used, as shown in Fig. 4 below, which replicates Fig. 2 a) in the manuscript with higher resolution (AOD bin 0.005).

The large spread of the data reflects the fact that the ACOS algorithm is not an aerosol retrieval algorithm. This was added to the text.

Most of the statistics, such as the correlation coefficient, averages, linear fits, and fraction of data in different quarters are calculated from the original data points, not from the aggregated grid cells. Only the average OCO-2 AOD values for each MODIS AOD bin (dashed red line) is calculated using binned data.

[Figure]

Figure 4. Density scatter plot of collocated AOD from OCO-2 and MODIS; same as Figure 2 a) in the manuscript, but using higher resolution (AOD bin 0.005, whereas Fig. 2 a has bin size 0.02). Grid cells with less than 10 points are not shown.

L293: We have added the text "The large spread of the data reflects the fact that the ACOS algorithm is not optimized for AOD retrieval, as discussed above. Considering this, the obtained correlation with MODIS AOD can be considered acceptable."

*10) L236: Do you have references or analyzed results? I agree with the cloud contamination. However, the effect of ice aerosol component is not clear. Please include some back-up result.*

**Reply:** As discussed in Section 2.1, the OCO-2 total AOD component includes contributions from water and ice particles. Preliminary comparison indicates that these scatterer types are more dominant in the low MODIS AOD/high OCO-2 AOD part of the AOD matrix, as indicated by Fig. 5 below. However, confirmation of this would require a more detailed look in the OCO-2 retrieval algorithm, which is beyond the scope of this work.

We have added a note to the text.

[Figure]

Figure 5. Contributions of water and ice components to the OCO-2 total AOD.

L283: We have added the text "These two AOD components are included in the ACOS retrieval to account for possible residual cloud contamination, while the MODIS aerosol retrieval does not have corresponding elements. Preliminary study shows elevated water and ice AOD values at low MODIS AOD values, but a more detailed study, beyond the scope of this work, would be required to confirm this."

*11) L240: Based on the statistical results and figures, this paragraph is not clear. The statistical score is possible to change due to the large number of data under low AOD grids. Statistical score change is not efficient in explaining the quality change of datasets.*

**Reply:** We have reformulated the paragraph. The smaller slope of the linear fit and the lower correlation coefficient are a natural consequence of removing all data points with OCO-2 AOD over 0.2, while a large fraction of the high MODIS AOD pixels remain in the dataset after applying the OCO-2 quality filter. The slope in the quality filtered dataset is rather meaningless, and we merely wanted to point out that for the AOD comparison we need to use the full dataset. However, we think that showing both panels in Fig. 2 helps the reader to understand the effect of OCO-2 quality filtering, which is used in the XCO2 retrieval.

L291: We have replaced the text "Pearson correlation coefficient for the unfiltered data is 0.60, reducing to 0.52 for the filtered data, which indicates that the sampling is biased for the quality filtered data (higher MODIS AOD values remain in the collocated data set). Note that in the collocated dataset the MODIS data is often the limiting factor (Table A1), already removing data over bright surfaces and in proximity of clouds. Applying the OCO-2 quality filter further reduces the collocated data to 56% of the original collocated data points. Most, but not all, of this reduction can be contributed to removing the high AOD cases."

 by

"The Pearson correlation coefficient for the unfiltered data is 0.60, reducing to 0.52 for the data filtered with the OCO-2 quality filter. The large spread of the data reflects the fact that the ACOS algorithm is not optimized for AOD retrieval, as discussed above. Considering this, the obtained correlation with MODIS AOD can be considered acceptable. Note that in the collocated dataset the MODIS data is often the limiting factor (Table A1), already removing data over bright surfaces and in proximity of clouds. Applying the OCO-2 quality filter further reduces the collocated data to 56% of the original collocated data points. Note that only 15% of the original data is removed by the total AOD threshold of 0.2, while 29% are removed by other quality tests. The lower correlation coefficient of the quality filtered dataset reflects the imbalance between OCO-2 and MODIS in the AOD distribution of data points removed by the OCO-2 quality filter."

*12) L293: Do you have reference?*

**Reply:** Lines 292-294 in the manuscript: "Figure 4 b) shows the correlation between MODIS AOD and OCO-2 XCO 2 for $1° × 1°$ grid cells. We see particularly high correlation values for the Sahel region, parts of South-East Asia, and Western USA."

We are not sure what the Referee means here. On line 293 we simply describe the results shown in Fig 4 b), without any reference to previous literature.

*13) Figure 5: Showing the number of data in each bin as adding figure.*

**Reply:** The number of data for each AOD bin is shown in Fig. 2 in the manuscript (on logarithmic scale). We have added a note on this to the text describing Fig. 5.

L365: We have added the note: "(see Fig. 2 for the number of data)".

*14) Section 4.4: So, from this section, does the author think that the AOD affects the XCO2 retrieval? How to be quantitatively separate the effects between the AOD effect and real XCO2 enhancement?*

**Reply:** As discussed in Section 4.3, it is difficult to disentangle the effects of real correlation between aerosols and CO2, and an aerosol bias in the XCO2 retrieval. Our conclusion is that there is a small bias in satellite XCO2 caused by aerosols, such that in heavy aerosol conditions the XCO2 is biased low. This acts to partly mask the true correlation between AOD and XCO2, but the bias has considerably smaller magnitude than the observed co-emission. We have reformulated the text to make this clearer.

(See point 6 above for the description of changes made in the text.)

*15) Section 5: I am confused about whether the AOD threshold change is acceptable.*
*For focusing on the comparison between XCO2 and MODIS AOD, the moderate AOD condition will make it possible to estimate the accurate XCO2 value. However, it is just the data based on the AOD from MODIS.*

*The AOD difference between OCO2 and MODIS is partially due to the AOD retrieval limitation by the OCO2.*

*In this case, high AOD conditions from OCO2 have high uncertainty. From this study, is this case can be clarified?*

**Reply:** Clearly the discussion in section 4.5 was not sufficiently clear, as pointed out by all referees. The purpose here was to demonstrate that the higher AOD threshold of 0.5, planned to be used in the coming CO2M mission (which will be better equipped to deal with higher aerosol concentrations), will bring considerable enhancement to the coverage. This is important in the light of the discovered co-emission of aerosols and CO2, since otherwise omitting high AOD areas might lead to biases in the XCO2 (source) data. Note that in Section 4.5 the MODIS AOD threshold is applied to data which has already been filtered using the OCO-2 quality filtering. We are unable to, and do not attempt to, judge if a higher (OCO-2) AOD threshold could be used with OCO-2 instrument.

We have extended the discussion in Section 4.5 to address the points raised by the referees and to clarify our message. Please see the more detailed answer in our reply to Referee #1.

(See our reply to Referee #1 for the changes made to the text.)

Referee #3

*This paper focuses on the collocation of OCO-2 and MODIS data, and analyzes the relationship between CO2 retrievals from OCO-2 and AOD retrievals from both OCO-2 and MODIS. The authors demonstrate that errors in AOD retrievals affect XCO2 retrievals, and also show that excluding data points with moderate AOD (0.2-0.5) excludes many areas with high XCO2. As a result, the authors recommend relaxing the MODIS AOD threshold to 0.5.*

**Reply:** We thank the referee for the valuable comments. We have revised the manuscript accordingly, to make the analysis and results clearer.

We see that our main message in section 4.5 was not stated clearly enough. We do not actually recommend relaxing the AOD threshold to 0.5 for the (OCO-2) XCO2 retrievals; we simply wanted to test the effect of using the looser threshold (planned to be used with CO2M) on the coverage in different environments. We have modified the abstract and conclusions to emphasize our main results, and section 4.5 to clarify the purpose of the aerosol threshold exercise.

L20: Added sentence "This crucial for monitoring anthropogenic CO2 emission, considering the observed co-emission of aerosols and CO2."

L512-523: We changed

"There is also co-variability between the AOD difference and the retrieved XCO2 values, most strikingly visible for the unfiltered dataset: overestimation of AOD in the OCO-2 retrieval leads to lower XCO2 values, and underestimation of AOD leads to higher XCO2 values. We have also found evidence of real covariance of AOD and XCO2, which is partly masked by the aerosol effect on the XCO2 retrieval. This covariance is presumably at least in part due to co-emission of anthropogenic CO2 and aerosols."

to

"The observed difference depends on location and conditions, but on average OCO-2 tends to overestimate at low aerosol loads and underestimate at higher AODs. We have found evidence of covariance of AOD and XCO2, which is presumably at least in part due to co-emission of anthropogenic CO 2 and aerosols. There is also co-variability between the AOD difference and the retrieved XCO2 values, most strikingly visible for the unfiltered dataset: overestimation of AOD in the OCO-2 retrieval is observed at lower XCO2 values and underestimation of AOD at higher XCO2 values. Comparison with TCCON reveals a weak but statistically significant dependence of the XCO2 bias on the AOD, such that at high AOD OCO-2 tends to underestimate XCO2. This aerosol bias acts in the opposite direction than the observed covariance between AOD and XCO2, partly masking the correlation. However, disentangling the effects of real covariance and aerosol bias is not straightforward, and we were not able to directly connect the observed AOD difference between MODIS and OCO-2 to the XCO 2 difference observed between TCCON and OCO-2."

L527: Added sentence: "In the light of the correlation found between AOD and XCO2, the AOD threshold affects also the average XCO2 values of the quality filtered data."

*Major comments*

*The authors note (in lines 181-185) that there is low correlation between MODIS AOD and OCO-2 AOD in Australia, the Sahel, the Western US, and Central Asia, using MODIS Dark Target observations. However, these areas seem to include bright land surfaces like large deserts and snowy mountain ranges, so perhaps using MODIS Deep Blue would be more appropriate for such areas. The Western US and Sahel also show high correlation between XCO2 and MODIS AOD in Figure 4b. Can this be explained by poor quality MODIS Dark Target observations? Would using MODIS Deep Blue for these areas change the analysis?*

**Reply:** Indeed, the areas of low correlation listed in the manuscript all have bright surfaces, as noted by the Referee. This might be one contributing factor to the observed correlations. We have added this to the text. However, as for MODIS Deep Blue algorithm, the same areas have low AOD correlations, as shown in Fig. A2 c). We have added a more explicit reference to this in the text.

L189: We have changed

"High AOD areas due to anthropogenic aerosol emissions are seen in particular in parts of Asia, biomass burning aerosols increase AOD in central Africa and South-East Asia, and elevated aerosol loads due to dust are seen over various desert areas around the globe."

to

"High AOD areas due to anthropogenic aerosol emissions are seen in particular in parts of Asia and elevated aerosol loads due to dust are seen over various desert areas around the globe."

*Section 4.5: The authors show that increasing the AOD threshold to 0.5 will increase the fraction of acceptable data, but do not show or discuss how this will affect the quality of XCO2 retrievals. It seems that improving the quality of XCO2 retrievals at higher aerosol loads remains an open challenge -- the authors should state this explicitly.*

**Reply:** Clearly the discussion in section 4.5 was not sufficiently clear, as pointed out by all referees. The idea here is that the coming CO2M mission will have a dedicated aerosol instrument - Multi-Angle Polarimeter (MAP) – which is expected to allow XCO2 retrieval at heavier aerosol conditions, with AOD threshold of 0.5. Here we use the MODIS AOD data collocated with good quality OCO-2 retrievals, which includes high MODIS AOD pixels (although the OCO-2 quality filter including an AOD threshold of 0.2 has been applied). With this collocated data set, we can test what is the effect of relaxing MODIS AOD threshold from 0.2 to 0.5.

We have clarified the text in section 4.5 to bring these points explicitly out. Please see our reply to Referee #1 for a more detailed answer.

*Minor comments/technical corrections*

*Line 299: Change (A3) to (see Fig. A3)*

**Reply:** The reference was incomplete. We have changed this to "(see Table A3)".

---

## Author Response (AR2)

Dear Editor,

In this response we reply to the comments by Referee #3 and Referee #4 for the second revision of our manuscript "A global perspective on CO2 satellite observations in high AOD conditions".

**Report #2 by Referee #3**

*The paper compiles and analyzes a dataset of XCO2 and aerosol observations from MODIS and OCO-2. The authors specifically note a correlation between AOD and XCO2, investigate the effects of relaxing the MODIS AOD threshold to 0.5, and find differences between MODIS and OCO-2 AOD retrievals as well as issues with the OCO-2 quality flag.*

**Reply:** We thank the referee for going through the of the manuscript again after revision, and for the useful comments. We have revised the manuscript accordingly and answer the questions below.

*General comments:*

*The paper is very long, and it is confusing to flip between the text of the paper and figures which only appear in the appendix. I recommend changing this so that all figures which are mentioned in the main text also appear in the main text. This may require making the paper more concise or removing some supplementary figures. I would consider cutting section 4.4, as I think the results there are relatively minor.*

**Reply:** We admit that the manuscript is long and has many figures. We tried to make the main text part of the original manuscript shorter and easier to read by placing some of the figures we found less crucial to the Appendix. The choice of most relevant figures for the main text was not easy, and as the Referee pointed out, this forces the interested reader to flip back and forth between the Appendix and the main text.

We agree that the results in Section 4.4 (Temporal and spatial dependence) are less crucial for the main points in our work, and could be removed to shorten the manuscript. We have now removed Section 4.4 and only briefly summarize the results in section 4.3. We have removed Fig. 8 from the main text and Figs. A9 and A10 from the Appendix. We also removed Tables A2-A3 from the Appendix, and reduced Tables A4 and A5 to the four main subsets of the collocated dataset. We have also removed from the text the references and discussion related to the removed figures and tables.

We have moved Figs. A3, A5 and A8 to the main text. We consider that Figures A1, A2, A4, A6, and A7 are less crucial and are only briefly mentioned in the text and can remain in the Appendix, to not lengthen the main text section. Figures A2 and A6 simply repeat Figures 1 and 5 from the main text with MODIS DB data. Figures A1, A4 and A7 are more technical, and not strictly necessary for following the main text, but contain some detail for the interested reader.

*The paper contains several different analyses (in sections 4.1-4.5) which are not necessarily obviously related. For example, I do not understand how the analysis in section 4.5 on coverage improvement resulting from relaxing the AOD threshold relates to the analysis in section 4.1 on the relationship between OCO-2 AOD retrievals and MODIS AOD retrievals. Perhaps the authors can edit the abstract and the conclusion to draw more clear connections between the different analyses, and can discuss how the different findings relate to each other. I think this would enhance the paper and make the impact of the work more clear to readers.*

**Reply:** We thank the Referee for pointing this out, and we now try to explain the connections in more detail both in the abstract and in the conclusions. We have also rewritten the introduction to Section 4 to better explain the connection between the subsections. Furthermore, we have relabeled sections 4.1 and 4.2 to better reflect the contents.

In Sections 4.1 we showed that there is a spatially varying AOD difference between OCO-2 and MODIS, and the difference is largest in areas of constant high aerosol loading. This means that a possible XCO2 bias related to the aerosol treatment in the ACOS retrieval could have different magnitude at different parts of the globe, which might cause uncertainties in emission estimates employing satellite data.

In section 4.2 we show that the OCO-2 quality filtering misses some high AOD cases, which remain in the so-called 'good quality' dataset, and this happens more frequently in areas of high anthropogenic emissions.

In section 4.3 we showed that there is a correlation between AOD and the XCO2 retrievals of OCO-2 (either a real correlation or a retrieval bias). With the TCCON comparison we then showed that this correlation is real, and not due to an aerosol related retrieval artefact. Thus, when the OCO-2 retrievals are limited by an AOD threshold, this causes a sampling bias, where the removal of high AOD areas leads to removal of high XCO2 values. Hence, it is important to estimate in Section 4.5 how much the sampling bias could be mitigated by relaxing the AOD threshold, especially in the urban areas which are critical in monitoring the anthropogenic CO2 emissions.

In section 4.3 we also found that XCO2 is biased low at high AOD. Unfortunately, the connection between the aerosol bias observed in sections 4.1 and 4.2 and the XCO2 bias observed in section 4.3 proved to be complicated, as the other retrieval parameters in the full physics retrieval can compensate for the errors in AOD, and we were unable to draw a direct connection.

(As the changes to the text in the abstract, in the introduction to section 4, and in the conclusions are rather large, we do not repeat them here but refer to the revised track-changed manuscript.)

**Report #3 by Referee #4**

*Review of "A global perspective on CO2 satellite observations in high AOD conditions" by Virtanen et al.*

*I was requested to review the revised version of the manuscript. In this manuscript, Virtanen et al. present an analysis of the collocated MODIS and OCO-2 satellite observations. The paper addresses an important question within the scope of AMT. One key conclusion is that there is a considerable difference between MODIS and OCO-2 AOD, which affects the XCO2 retrieval. I commend the authors for using several large datasets. I have some major questions related to your analysis. Therefore, the authors should address these questions before the manuscript can be published. For the following comments, please refer the page and line numbers to the track-changed manuscript.*

**Reply:** We thank the Referee for reviewing the revised manuscript and for the valuable comments. We agree that the comments about the quality of MODIS aerosol product as a reference data raised by the Referee are important. After carefully considering the comments, reviewing related literature, and making some additional analyses, we find that we can address these points without performing a full reprocessing of the global five year dataset with the MODIS 3 km aerosol product. We also

argue that since the MODIS aerosol product has already been extensively validated in the research literature, there is little benefit of repeating this considerable effort in this work. However, we readily admit that these points need to be better addressed in the manuscript. In the detailed responses below we show our additional analyses and explain how we take these concerns better into account in the revised version of the manuscript.

*Major comments:*

*1. If I understand it correctly, the authors perform the comparison between the MODIS Aqua 10 km product and OCO-2 AOD 1 km x 2 km product. In section 4.1, the authors note that it may affect the comparison because of the different spatial resolutions but does not consider how the spatial representativeness issue contributes to the large difference between the two products. And hence I doubt that the ACOS retrieval is solely responsible for the difference.*

*I suggest that you may minimize the representativeness issue by using the MODIS Aqua 3 km product (MYD04_3K) and possibly 2x2 OCO-2 AOD pixels (~2 km x ~4km).*

**Reply:**
We argue that the spatial variability of ambient aerosol fields is typically relatively smooth on the length scales considered, so that the choice of MODIS aerosol product should have minimal impact on the global scale. There is a wealth of literature on comparison of the MODIS 10 km and 3 km products (e.g. Gupta et al. (2018), He et al. (2017)). The general conclusion from this literature is that the two products perform rather similarly, on the global scale, in comparison with AERONET. This supports our view that the comparison to the finer spatial scale OCO-2 product should not depend much on the MODIS product resolution on the global scale.

In our analysis we heavily aggregate the individual collocated data points. In the AOD matrix plots we aggregate the data to AOD bins of width 0.02, with dozens of data points in each bin. In the global maps, we aggregate the data to 1x1 degree grid cells with thousands of data points in each grid cell. We argue that this aggregation further reduces any effect the MODIS product resolution might have on the results.

Furthermore, as we emphasize in the manuscript, OCO-2 does not provide a specific aerosol product. The total AOD value given in the OCO-2 product is just one of the dozens of parameters in the full-physics retrieval, and the values are largely dependent on the large-scale aerosol climatology values used as priors. Again, we do not expect to get a one-to-one correspondence with high spatial resolution reference aerosol data but are interested in the larger scale statistics.

We certainly acknowledge that the MODIS 3 km product can be useful for individual overpasses and e.g. for thick aerosol plumes. But as mentioned in the manuscript, detailed case studies are beyond the scope of this work, and we believe that the effect of the MODIS aerosol product resolution and sampling methods would better fit such work. We think that the cited literature supports this conclusion, in that the differences between the two MODIS products show up on a smaller scale, but are smoothed out in larger data sets. Similarly, we expect that averaging the OCO-2 data to 2x2 pixels would make a little difference in the end.

In order to verify our understanding that the choice of the MODIS product makes little difference on the global scale, we have processed a limited number of MODIS 3 km aerosol data (MYD04_3K). The land-global comparison of MODIS 3 km product with OCO-2 for the year 2018 shows minor differences compared to the similar comparison using the 10 km product (see Fig. R1 below). Hence, we are confident that our original analysis sufficiently covers the appropriate MODIS information.

[Figure]

Figure R1. Difference between MODIS 10 km (left) and 3 km (right) aerosol products in comparison with OCO-2 for one-year land-global dataset. The 3 km product provides more matches with OCO-2 but the main conclusions do not change.

Reprocessing the full land-global 5-year dataset with the MODIS 3 km product (which is larger in size compared to the 10 km product due to the increased number of data points) would require considerable resources, and we expect that the resulting difference compared to the existing work would be minimal.

We have now further elaborated our choice of the reference satellite aerosol dataset in the manuscript (Section 2.2), and the benefits of using higher resolution products in more detailed case studies. We also briefly discuss the results of the additional analyses using the 3 km product in Section 4.1. (Due to the request by other Referee to make the manuscript more concise, we do not include additional figures in the manuscript).

Added text, L89 (line numbers refer to the new track-changed manuscript):

> "Collocation with the higher spatial resolution MODIS 3 km aerosol product (MYD04_3K; Remer et al. (2013)) was tested for one year (2018). The results did not differ significantly from the corresponding subset when using the 10 km DT product (not shown). Due to the considerably larger computational burden of the 3 km data, the full dataset was processed only with the 10 km product. Previous studies have shown that the 3 km and 10 km products perform very similarly on the global scale (Gupta et al., 2018; He et al., 2017). For more detailed case studies the use of MODIS 3 km product could be beneficial, but that is beyond to scope of this exercise."

Added text, L220:

> "A limited collocation test made with MODIS 3 km aerosol product for the year 2018 shows slightly enhanced coverage but otherwise very similar AOD patterns as the 10 km DT product."

*2. The MODIS retrieval can also contribute to the AOD difference between MODIS and OCO-2. While the comparison between OCO-2 and MODIS AODs is very detailed, it is not very useful information unless you know which one is better with respect to AERONET (sort of a ground truth). I agree that the MODIS Angstrom exponent can be uncertain. But alternatively you can 1) compare MODIS AOD at 550 nm with AERONET and 2) compare OCO-2 AOD at 755 with AERONET using AERONET Angstrom exponents. You have compared OCO-2 with AERONET in Figure A5. I suggest that you should do it again using MODIS.*

**Reply:**
We acknowledge that the MODIS aerosol product has uncertainties which affect the comparison. However, we find that the MODIS product is probably the most extensively validated satellite aerosol product there is [e.g. Levy et al. (2013), Sayer et al. (2014), Wei et al. (2019)], and we see little benefit in repeating the validation against AERONET in this manuscript. Clearly, we have failed to discuss the MODIS aerosol product quality in the manuscript, and this has now been improved in the latest revision.

The validation studies typically show a correlation coefficient R ~ 0.9 and small bias for MODIS against AERONET, while for OCO-2 we find R ~ 0.7 and a considerable low bias at large AERONET AOD values. We also find a striking similarity in the comparisons of OCO-2 AOD data to AERONET and to MODIS data, in that OCO-2 tends to have a low bias at high AOD values. This supports the view that the MODIS product can be used as a global reference for AOD. Here we want to once again emphasize that we are not attempting to validate the OCO-2 total AOD component as if it was a dedicated aerosol product; the OCO-2 total AOD is one of the many retrieval parameters of the full-physics retrieval, and we are mainly interested in the large scale statistics when comparing with the MODIS aerosol product.

As pointed out by the Referee, we evaluate OCO-2 aerosol data against AERONET. To further address the Referee's concerns, we have performed a separate validation of the MODIS part of the collocated dataset against AERONET (see Fig. R2 below). This differs from the typical validation in that the sampling is not optimal for MODIS but limited to the pixels collocated with OCO-2. As expected, this sampling leads to slightly reduced validation metrics against AERONET (R ~ 0.8, small bias), but still better than for OCO-2. We note that while repeating the validation of the full MODIS dataset would be laborious, using the collocated dataset for this limited exercise was straightforward and better fits the scope of this work.

We now discuss the MODIS aerosol product quality and suitability for global AOD reference for this study in more detail in the manuscript. We also draw attention to the similarity of the OCO-2 comparison against AERONET and against MODIS, and discuss the results of the limited validation of the MODIS component of the collocated dataset (Section 4.1). Since we were requested to make the manuscript more concise, we will not use additional figures in the manuscript.

Added text L256:
> "The MODIS aerosol products have been extensively validated, with a typical correlation coefficient R∼0.9 against AERONET (Levy et al., 2013; Sayer et al., 2014; Wei et al., 2019). We do not repeat the MODIS AOD product validation against AERONET in this work, but we have compared the MODIS part of the collocated OCO-2/MODIS dataset to AERONET with similar sampling as used for OCO-2. This differs from the typical validation in that the sampling is not optimal for MODIS but limited to the pixels collocated with OCO-2. As expected, this sampling leads to slightly reduced validation metrics against AERONET (R∼0.8, small bias), but the metrics are still better than for OCO-2. Hence, we

are confident that although MODIS AOD product certainly has higher uncertainty than AERONET, it helps to the extend the evaluation of OCO-2 AOD to the global scale."

[Figure]

Figure R2. Comparison of the MODIS AOD from the collocated MODIS/OCO-2 dataset to AERONET at 550 nm. Collocated MODIS data is limited to the narrow OCO-2 swath. Collocation distance is 0.125º around the AERONET site and time window is ±1 h.

Regarding the wavelengths used in the comparison with AERONET, we used the approach of averaging two wavelengths for simplicity in Fig. A5. As discussed in the revised manuscript and in our reply to Referee #2 in the first revision of the manuscript (https://doi.org/10.5194/amt-2024-77-AC2), we have repeated the AERONET comparison with a subset of data using the Ångström exponent from AERONET, and showed that this has a small effect on the results. This is also discussed in the revised manuscript (Section 4.1, L251 in the new track-change manuscript). In Fig. R2 we used AERONET Ångström exponent to scale the AERONET measurements to 550 nm.

*3. MODIS quality flag can have values from 0 to 3 (best quality). While you remove the poor quality pixels (flag = 0), does it matter if you only compare with pixels with best quality.*

**Reply:**
We see that the discussion concerning the MODIS quality flag was quite brief in Section 2.2, and we have now extended it. The choice of quality flag is a trade-off between data quality and coverage (Fig. R3 below). While we concentrated on the OCO-2 quality flag, we have also tested the effect of using different MODIS quality flags in the filtering. We find that applying more strict filtering on the MODIS data reduces the number of data, reduces the average MODIS AOD by 0.02, and slightly improves the validation against AERONET, but does not significantly affect the comparison between OCO-2 and MODIS too much (see Figs. R3 and R4 below).

[Figure]

Figure R3. Effect of MODIS quality flag on the coverage of the collocated dataset. Left: only worst pixels (qf=0) removed. Right: only best pixels (qf=3) kept. OCO-2 quality filtering has been applied.

[Figure]

Figure R4. The effect of MODIS quality flag on the collocated data. On left, we have removed only the lowest quality MODIS pixels (qf=0), and on the right hand panel we have kept only the best quality MODIS pixels (qf=3). OCO-2 quality filtering has not been applied.

Since we are interested in the performance of the OCO-2 quality filtering, we did not want to limit the MODIS data too much. Also, we found that more strict quality filtering tends to remove a larger fraction of pixels from the urban areas, which are of specific interest in this study. We now discuss the effect of MODIS quality flag in more detail in the manuscript (Section 2.2).

Added text L101:
> "We also tested using more stringent quality filtering, keeping only the best quality MODIS data (quality flag 3). Although this reduced the number of matches with OCO-2 by nearly 30% and reduced the global average AOD by 0.02, it did not affect the conclusions of our work."

*Minor comments:*

*P4, L106: What does 0.01-0.02 represent? Absolute deviation or standard deviation or something else?*
**Reply:**
According to Eck et al. 1999 this is the estimate total absolute uncertainty in AOD (dimensionless). This is now clarified in the manuscript.

Revised text L115:
> "The AERONET sunphotometer measurements are routinely used as reference measurements for satellite aerosol retrievals due to their high accuracy ($\sim$ absolute error in AOD of the order 0.01-0.02, Eck et al. (1999); Sinyuk et al. (2020))."

*P5, L132: What is GGG2020? Please define or explain it.*
**Reply:**
GGG2020 is the name of the latest version of the retrieval algorithm used in the TCCON retrievals (Laughner et al., 2024, https://doi.org/10.5194/essd-16-2197-2024). We have added the explanation and reference to the manuscript.
Revised text L137:

> "The effect of different prior profiles in OCO-2 v10 and TCCON retrieval algorithm version GGG2020 (Laughner et al., 2024) was taken into account by adjusting the OCO-2 XCO2 value, following Mendonca et al. (2021)."

*Figure 2: x-av, y-av -> Should it be x-axis and y-axis?*
**Reply:**
In Figs. 2 (as well as in Figs 5, 6, A4, A6, and A8) the text insets on the bottom right 'x-av' and 'y-av' refer to the average value of the quantities shown on the x and y-axis, respectively (here 'av' is short for average). This is explained in the caption of Fig. 2.

*P21, L470: seasons -> season*
**Reply:**
This was a typo. The entire Section 4.4 has now been removed as requested by Referee #3.

---

## Author Response (AR3)

Dear Editor,

In this response we reply to the final comments by Referee #4 for the latest revision of our manuscript "A global perspective on CO2 satellite observations in high AOD conditions".

**Report #1 by Referee #4**

*The authors have adequately addressed to most of my questions. I now have only few minor comments. After that, I recommend the manuscript for publication.*

**Reply:** We thank the referee for going through the of the manuscript again after the latest revision, and for spotting several typos and other deficiencies in the text. We have revised the manuscript accordingly and answer the comments below.

*Minor comments:*

*P2, L59: we will also -> we also*
*P3, L90: to -> the*
*P10, L251: helps to the extend -> helps to extend*
*P11, L284: using e.g. density scatter plots -> using, for example, density scatter plots.*
*P13, L326: resulst -> results*

**Reply:** These five points have been changed as suggested.

*P21, L451: what relationship?*

**Reply:** We refer to the observed linear relationship. We have changed the text from "we find that there is a relationship between OCO-2 XCO2 and MODIS AOD" to "we find that there is a linear relationship between OCO-2 XCO2 and MODIS AOD".

*Figure 1(c) title: btw -> between*

**Reply:** We have changed the image titles in Figs. 1(c) and 6(b) as suggested.

*Figure 2: Both the solid and dotted lines are too thin. This also happens in some other figures which makes it hard to see.*

**Reply:** We now use thicker solid lines and have replaced the dotted lines indicating the interquartile range by shaded areas to make the figure clearer. We have also used thicker lines in Figs. 4, 8 and 10 to make them easier to see.